# Concept neurons in the human medial temporal lobe flexibly represent abstract relations between concepts

Marcel Bausch [1✉], Johannes Niediek [1], Thomas P. Reber [1,2], Sina Mackay[1], Jan Boström[3], Christian E. Elger [1] & Florian Mormann [1✉]

Concept neurons in the medial temporal lobe respond to semantic features of presented stimuli. Analyzing 61 concept neurons recorded from twelve patients who underwent surgery to treat epilepsy, we show that firing patterns of concept neurons encode relations between concepts during a picture comparison task. Thirty-three of these responded to non-preferred stimuli with a delayed but well-defined onset whenever the task required a comparison to a response-eliciting concept, but not otherwise. Supporting recent theories of working memory, concept neurons increased firing whenever attention was directed towards this concept and could be reactivated after complete activity silence. Population cross-correlations of pairs of concept neurons exhibited order-dependent asymmetric peaks specifically when their response-eliciting concepts were to be compared. Our data are consistent with synaptic mechanisms that support reinstatement of concepts and their relations after activity silence, flexibly induced through task-specific sequential activation. This way arbitrary contents of experience could become interconnected in both working and long-term memory.

[1] Department of Epileptology, University of Bonn Medical Center, Bonn 53127, Germany. [2] Faculty of Psychology, UniDistance Suisse, Brig, Switzerland. [3] Department of Neurosurgery, University of Bonn Medical Center, Bonn 53127, Germany. ✉email: marcel.bausch.ukb@gmail.com; florian.mormann@ukbonn.de

oncept neurons in the human medial temporal lobe (MTL) respond to semantic features of presented stimuli and are thought to represent elements of experience[1]. The hippocampus encodes temporal[2] as well as abstract[3] relations among elements of experience across spatiotemporal gaps[4]. These representations can be modulated by cognitive insight[5]. Furthermore, hippocampal lesions have been associated with relational memory deficits, both in tasks involving long-term[6] and even working memory[7,8], particularly under high memory load, interference by other memory items, and longer time intervals[7–9]. While persistence of firing has been proposed to represent currently attended memory items, synaptic mechanisms could account for the recovery of firing after activity silence in both working and long-term memory[10]. Visually selective neurons in the human MTL represent preferred visual stimuli in working memory through persistent firing[11–13] that predicts memory performance[11,12] and reflects associated stimuli[14–16]. How concept neurons dynamically encode conceptual relations and account for their storage, however, is not completely understood. Many instances in which a general concept occurred can be recalled with ease, and multiple episodes become associated through perceived or factual relations between their contents. Concept neurons are a prime candidate for such a dynamical linkage of memories through conceptual connections[17]. Here, we employed a picture comparison task to show that concept neurons encode abstract relations between concepts. Note that relations between everyday concepts are highly dynamic: they cannot be anticipated in advance, they change over time, and only some of them will be stored permanently into long-term memory. Working with human subjects, we had the unique opportunity to vary picture relations merely through abstract task instructions while preserving the same overall trial structure. Importantly, concept neuron activity reflected conceptual relations dynamically and only if the abstract task instruction meant that the relation was actually relevant. Firing increased whenever attention was directed toward the response-eliciting concept, even in anticipation of the next stimulus and after periods of complete activity silence. Trial-wise sequential firing patterns suggested a synaptic[18–20] and/or cell-intrinsic[21,22] storage mechanisms to account for the activity-silent retrieval of concepts and their relations.

## Results

### Concept neuron activity for various relations between concepts.

During 38 experimental sessions, we recorded from 2512 neurons in the amygdala, parahippocampal cortex, entorhinal cortex, and hippocampus of twelve neurosurgical epilepsy patients implanted with depth electrodes for pre-surgical evaluation. Patients were asked to compare either the semantic content ("Bigger?", "Last seen in real life?", "More expensive?" or"Older?" depending on the stimulus set, "Like better?") or non-semantic stimulus features ("Brighter (picture)?") of two pictures presented on a laptop screen (Fig. 1a). For this task we showed pairs from four pictures selected based on a previous screening procedure to maximize the number of responsive concept neurons. Each trial began with the presentation of one of five comparison questions (such as "Bigger?"), followed by a sequence of two of the four pictures. Subjects then chose the picture that answered the question (e.g., which depicted something bigger) and indicated whether it was shown first or second by pressing keys 1 or 2 on the keyboard. Afterward, two control conditions with identical trial structure were run. In contrast to the main condition, controls did not require a comparison of the pictures themselves, nor of their contents. During the no-comparison control condition subjects counted the number of fixation crosses displayed in red instead of white (Fig. 1b). During an additional question-comparison control condition, one

of the four picture concepts was mentioned in the question text and had to be compared to both subsequent pictures (Fig. 1c).

We defined visually selective neurons as neurons that respond to exactly one of the four pictures in the main experimental condition (bin-wise signed rank test with Simes' correction against baseline, alpha = $10^{-5}$, see Methods). Moreover, the response to the preferred picture had to markedly exceed those to non-preferred pictures in each experimental condition (Hedges' $g > 0.3$). Concept neurons were defined as visually selective neurons with stronger responses to those question instructions that contained the name of the preferred concept versus those that did not (Hedges' $g > 0.3$).

### Concept neurons indicate the presence of a relation to their preferred concept.

Overall we identified 128 visually selective units. Of these, 61 units qualified as concept neurons (Supplementary Fig. 1). Thirty-three of these concept neurons responded to non-preferred concepts with a delayed but well-defined onset whenever the task required a comparison to the response-eliciting concept (~400 ms later than the response to the preferred stimulus; Fig. 1d). Figure 1d–f shows an example of a concept neuron whose preferred concept was a tie. During the main-comparison condition when the tie was shown first, the neuron started responding to non-preferred pictures shown second, but with a delayed onset. We sought to determine whether these late responses depended on a mental comparison to the preferred concept and could hence be interpreted as relational reactivation responses. Neither control condition required a comparison between the concepts depicted in the pictures. Either no comparisons (no-comparison control) or comparisons of both picture concepts to a third concept mentioned in the question (question comparison) were required. Since no concepts were presented more than once within the same trial, a preferred first picture (tie) in the question-comparison condition meant that the second picture had to be compared to a non-preferred concept mentioned in the question. Consequently, relational responses during second non-preferred picture presentations were abolished in both control conditions (Fig. 1d, no comparison and question comparison). Strikingly, however, during the question-comparison control, approximately one fourth of concept neurons responded not only to the questions containing the preferred concept ("tie") but also exhibited relational responses during the presentation of both subsequent non-preferred pictures, again with a delayed onset (Fig. 1e). Relational responses were not present when neither the question nor the pictures referred to the preferred concept (Fig. 1f). The behavior observed at the level of individual units was well preserved when we analyzed the entire population of concept neurons (Fig. 2). Relational responses were associated with significantly increased normalized firing (cluster permutation test against zero) during second, non-preferred picture presentations following the preferred picture in the main experiment, but not in control conditions (Fig. 2a, see also Fig. 1d). These analyses confirm that reactivated activity occurred ~400 ms later than the response to the preferred stimulus and was absent in both control conditions. During the question-comparison control condition, approximately one third of the concept neurons responded not only to questions containing the preferred concept but also to both subsequent non-preferred pictures (Fig. 2b, top row). Responses were absent when neither the question nor the pictures depicted the preferred concept (Fig. 2b, bottom row). Finally, we tested whether response and reactivation patterns of concept neurons were consistent across experimental conditions. Factors predictive of reactivations in the question control condition were visualized as scatter plots of mean normalized activity for all 61 concept neurons (Fig. 2c).

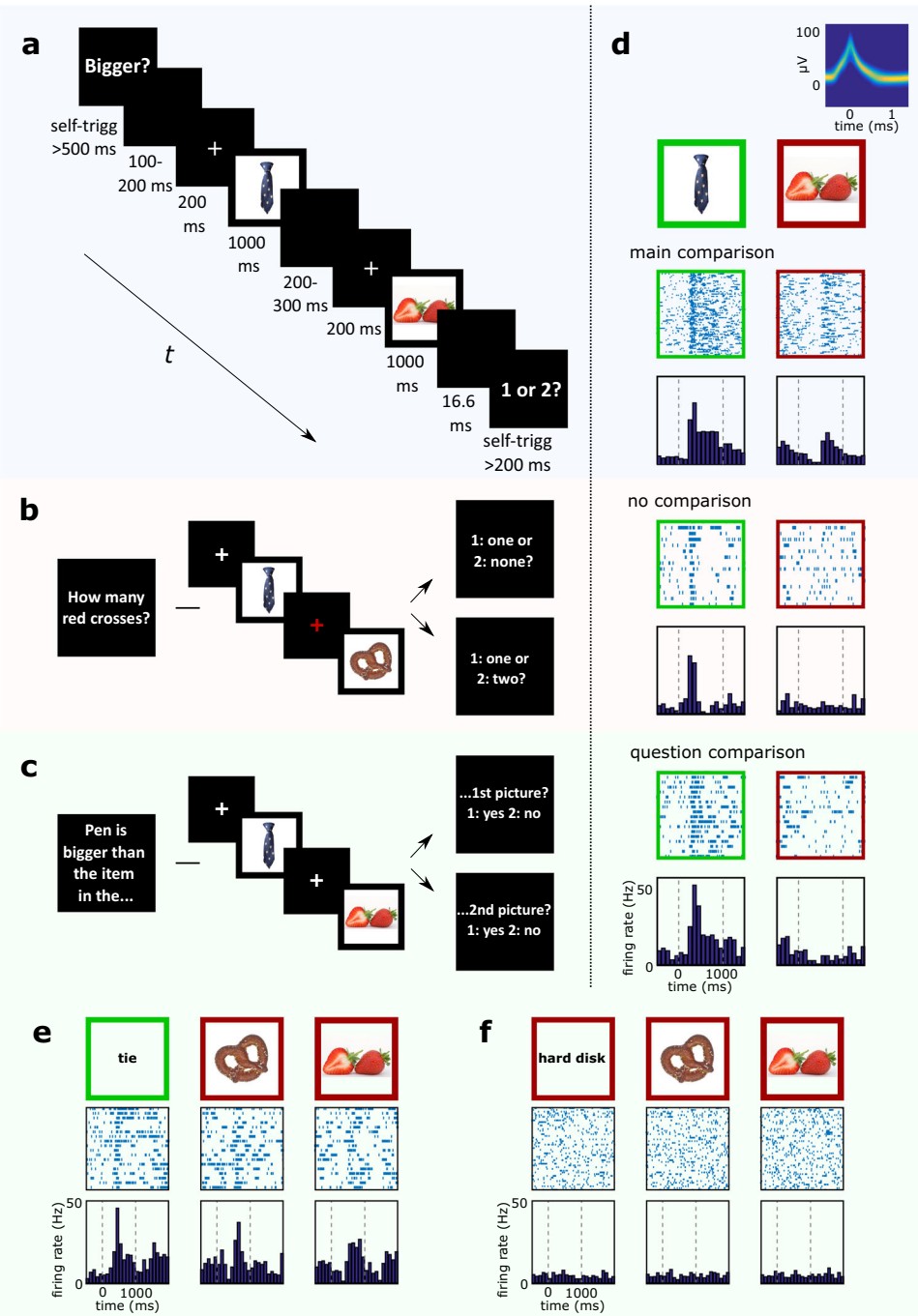

**Fig. 1 Concept neurons respond after presentation of otherwise non-preferred stimuli whenever a comparison to the previously shown preferred stimulus is required. a** Main-comparison condition (blue). Each trial contained one of five questions ("Bigger?", "Last seen in real life?", "More expensive?" or "Older?", "Like better?","Brighter?"), a sequence of two pictures (out of four), and an answer prompt displaying "1 or 2?". Subjects indicated the sequential position of the picture that best answered the question by pressing keys 1 or 2. Question and answer screens were self-triggered (self-trigg). Event durations are printed below. **b** No-comparison condition (red). Same structure as in a. Subjects had to count the number of red fixation crosses. Neither, one, or both of them could be red. **c** Question-comparison condition (green). Similar to a, but here the question also referred to one of the stimuli by text. Two of the three remaining pictures followed. Subjects had to compare both pictures to the stimulus named in the question. **d** Raster plots, histograms and density plot of one concept neuron from all three experimental conditions (**a–c**) during first and second picture presentations. Frame color indicates whether the stimulus was preferred (green) or non-preferred (red). This neuron responded not only to the preferred picture (tie) but became reactivated in response to any subsequent non-preferred picture (e.g., strawberry) whenever the task required a comparison to its preferred picture (**a**), but not otherwise (**b**, **c**). **e** Same neuron as in **d** during the question-comparison condition for trials in which the preferred concept (tie) was part of the question. A response to the question and to both subsequent non-preferred pictures was observed (pretzel and strawberry). **f** Same as in **e** for trials in which neither the question, nor the subsequent pictures contained the preferred concept. The neuron did not respond at all. Pictures of objects (**a–f**) were obtained from https://commons.wikimedia.org under the Creative Commons CC0 1.0 Universal Public Domain Dedication.

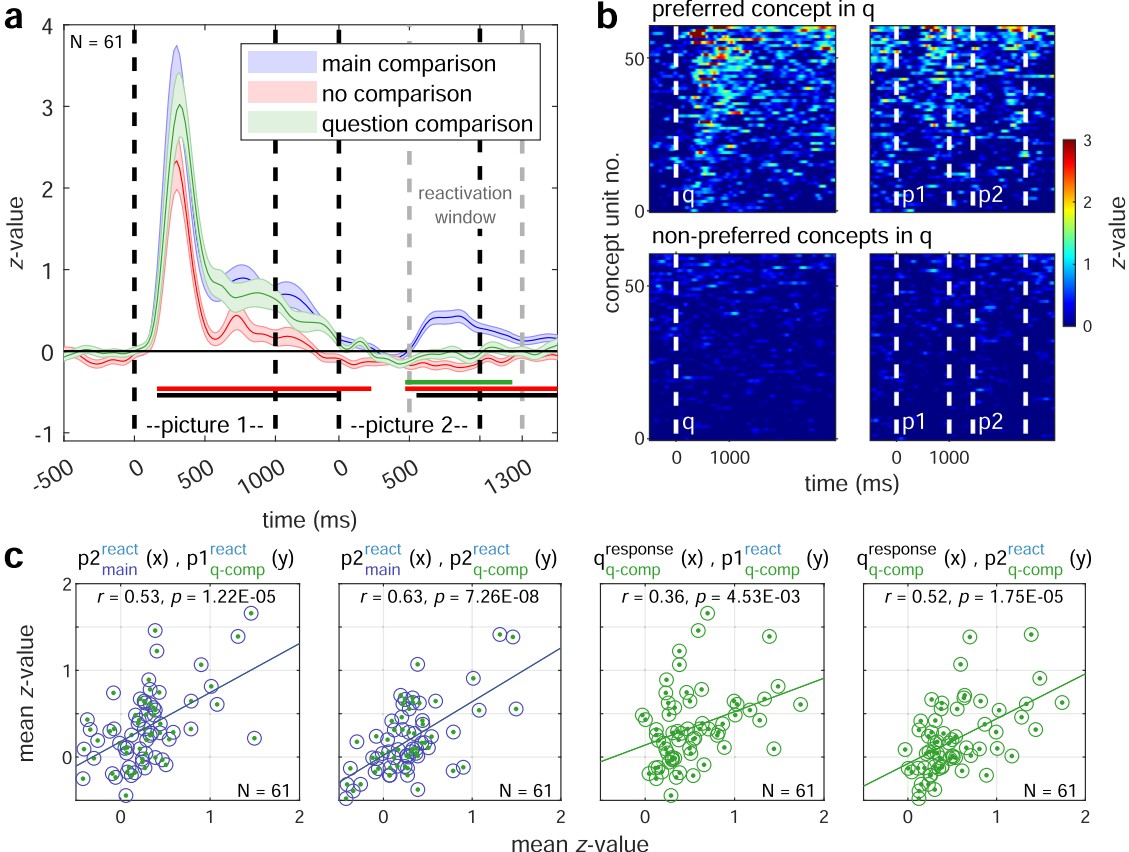

**Fig. 2 Relational responses of the entire population of concept neurons. a** Averaged normalized firing rates of concept neurons during both picture presentations in all three experimental conditions whenever the first picture represented the preferred concept. Data are expressed as mean ± SEM (solid lines and shaded areas). Picture on- and offsets are marked by black dashed lines. Relational responses occurred during the reactivation window (gray dashed lines) in the main-comparison condition (blue), but not in the no- or question-comparison control condition (red and green, respectively). Time periods of significant z-value differences between main and control conditions (same colors) or zero (black) are indicated by solid lines ($p < 0.01$; two-sided cluster permutation test). **b** Heat plot of mean z-values of all 61 concept neurons during the question-comparison condition sorted in descending order of activity. Dashed white lines denote onsets of different events (q: question, p1: picture 1, p2: picture 2). Top row: When the preferred concept was part of the question, responses to the question and following both non-preferred pictures were present. Bottom: When neither question nor pictures contained the preferred concept, no responses or reactivations occurred. **c** Scatter plot of mean z-values of all 61 concept neurons comparing responses and reactivations during different stimulus presentations (q: question, p1: picture 1, p2: picture 2) and experimental conditions. Subscripts indicate the experimental condition (main in blue: main experiment, q-comp in green: question comparison) while superscripts distinguish whether the preferred concept was depicted in the current (response: response trials 0–1000 ms after stimulus onset) or a preceding stimulus (react.: reactivation trials 500–1300 ms after stimulus onset). Pearson correlation strengths and p values (uncorrected) for each condition pair are shown at the top and visualized by regression lines. Left two subplots: Reactivations during the main experiment predict reactivations in the question control condition ($r > 0.5$, $p < 0.0005$). Right two subplots: Response strengths to questions containing the preferred concept predict reactivation strengths during both subsequent picture presentations ($r > 0.35$, $p < 0.005$). Source data are provided as a Source Data file.

Both reactivation response strength to second non-preferred pictures in the main condition (Fig. 2c, left two subplots) as well as preferred question response strength (Fig. 2c, right two subplots) predicted non-preferred picture reactivation strength in the question-comparison control. Specifically, Pearson correlations between reactivations in the main condition versus relational responses in the question-comparison condition to first ($r = 0.528$, $p = 1.224 \times 10^{-5}$) or second non-preferred pictures ($r = 0.625$, $p = 7.264 \times 10^{-8}$) as well as correlations between preferred question response strengths versus first ($r = 0.359$, $p = 4.534 \times 10^{-3}$) or second picture reactivations ($r = 0.520$, $p = 1.749 \times 10^{-5}$) in the question-comparison condition were highly significant.

**Pairwise relations are revealed by the activity of pairs of concept neurons.** We next asked whether preferred stimulus responses and reactivated activity of local pairs of concept neurons, i.e., neurons recorded from the same microwire bundle, could indicate the presence of concept relations on a trial-by-trial basis. Figure 3a shows two hippocampal concept neurons recorded on the same bundle of electrodes whenever both of their preferred concepts were presented consecutively. Normalized cross correlations were computed from their activity during first and second picture presentations for main comparison (blue) and both control trials (red). Cross-correlations only exhibited asymmetric peaks during second picture presentations and during the main comparison condition when a meaningful relation between two concepts had to be established. The same pattern was observed at the population level when all 22 local pairs of concept neurons were analyzed (Fig. 3b). Neither prominent peaks, nor significant differences between the main comparison and control conditions were detected during first picture presentations (Fig. 3b, top). During second picture presentations of

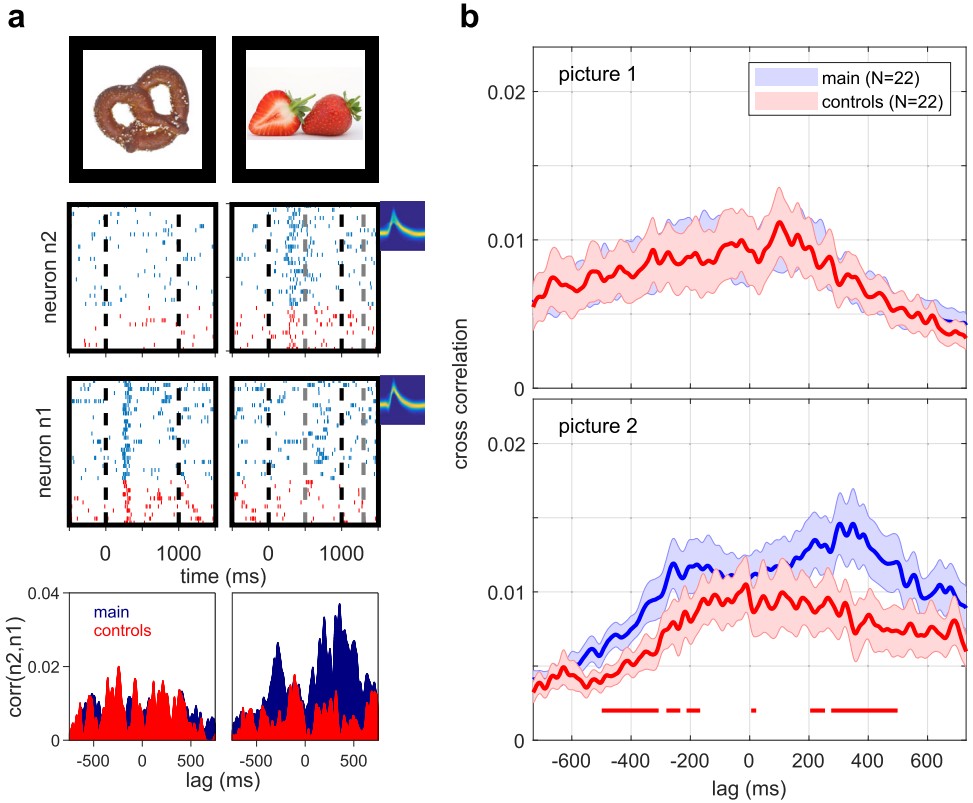

**Fig. 3 Firing sequences of local pairs of concept neurons reflect concept relations, potentially leading to synapse modification. a** Raster plots, normalized cross-correlograms and spike shapes of two concept neurons in the right hippocampus during the presentation of each neuron's preferred stimulus (pretzel for n1 and strawberry for n2). In the main condition (blue), but not during controls (red) neuron n1 responded to the non-preferred strawberry picture (relational response on lower right). Bottom: Trial-by-trial cross-correlograms between both neurons during each picture presentation with colors as above. Peaks around −250, and +350 ms were present only for second picture presentations in the main condition. **b** Population plots of trial-by-trial cross-correlograms between all pairs of concept neurons of the same brain region. Data are presented as mean values ± SEM (solid lines and shaded areas). Red horizontal lines indicate significant differences ($p < 0.01$) between the main-comparison condition and controls as quantified by a two-sided cluster permutation test. Pictures of objects (**a**) were obtained from https://commons.wikimedia.org under the Creative Commons CC0 1.0 Universal Public Domain Dedication. Source data are provided as a Source Data file.

the main condition, however, two prominent correlation peaks (around −250 ms and +350 ms) were found (Fig. 3b, bottom) and cross correlations differed significantly from control conditions on short (<25 ms) as well as longer (200–700 ms) timescales ($p < 0.01$; cluster permutation test, see Methods). The positive cross correlation peak corresponding to neuronal firing in the reverse order of the presentation of preferred pictures (i.e., a later reactivation of the response to the first picture) was most pronounced. Cross correlations for non-local pairs of concept neurons exhibited the same overall pattern (Supplementary Fig. 2a). In order to disentangle stimulus-induced correlations from potential interactions between concept neurons, we subtracted non-simultaneous cross-correlograms of consecutive trials (shift predictors) from simultaneous ones for non-local (Supplementary Fig. 2b) and local (Supplementary Fig. 2c) pairs of concept neurons. After correction, only local pairs still showed a cross correlation peak at around +300 ms during second-picture presentations that differed significantly between experimental conditions ($p < 0.05$; cluster permutation test).

**Representations of abstract relations are associated with local correlations.** While concept neurons were found in all brain regions of the medial temporal lobe (~2% of units; Fig. 4a, top), their proportion with respect to visually selective neurons was lowest in parahippocampal cortex. Furthermore, relational responses (reactivated neurons) were most frequent in areas

associated with declarative memory function, namely in amygdala, hippocampus and entorhinal cortex (~3% of units; Fig. 4a, bottom). Visual neurons, i.e., neurons that responded only to the presented stimulus, but not to the compared concepts (no relational responses), on the other hand, were most frequent in parahippocampal cortex. Normalized relational responses (500–1300 ms) only differed significantly from zero when the question required semantic processing of the pictures (semantic trials, $p = 1.74 \times 10^{-7}$; Wilcoxon signed-rank test, Fig. 4b). Perceptual processing (perceptual trials, i.e., "Brighter?") for concept neurons or visual selectivity alone (visual neurons, both perceptual and semantic trials) was associated with significantly lower normalized activity (all three $p < 0.005$; Mann–Whitney $U$ test) not significantly different from zero (Wilcoxon signed-rank test). Moreover, concept neurons responded more strongly to their preferred concept in second versus first picture positions, particularly during the early response phase immediately preceding the period of relational responses (reactivation window; Fig. 5a, $p < 0.01$; cluster permutation test) and even before the onset of the second picture (anticipation window; Fig. 5a). Despite being more pronounced in semantic trials, positional response strength differences did not differ significantly from perceptual trials (Supplementary Fig. 3a; $p = 0.13$; Wilcoxon signed-rank test). In order to capture potential interactions between pairs of concept neurons, we computed their pairwise correlations. For each pair, the response strength of one neuron (n2) to its preferred concept

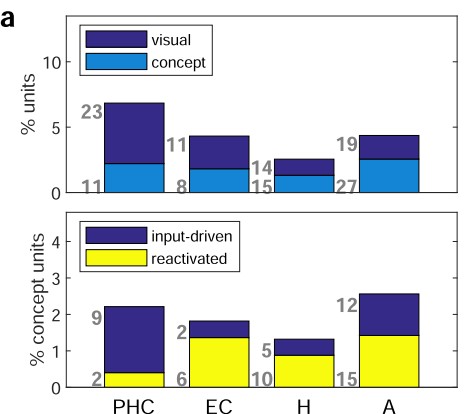
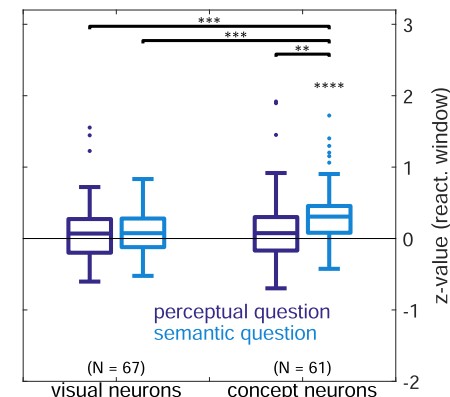

**Fig. 4 Relational responses occur in brain regions associated with episodic memory and require conceptual processing. a** Relative frequency of each neuron type expressed as percentage of recorded units for different brain regions (PHC parahippocampal cortex, EC entorhinal cortex, H hippocampus, A amygdala). Top: Visually selective neurons are divided into concept and non-concept (visually selective) neurons. Bottom: Concept neurons are split into reactivated neurons with relational responses and remaining input-driven neurons. **b** Boxplots of z-values (Q1, median, Q3; whisker: points within ±1.5 IQR) of average normalized relational response activity during the reactivation window of the main condition for merely visually selective versus actual concept neurons. Trials requiring semantic processing ("Bigger?", "Last seen in real life?", "More expensive/ Older?") are colored in light blue, those depicting the perceptual question ("Brighter?") in dark blue. Only relational responses of concept neurons during semantic questions deviated significantly from baseline ($p = 1.74 \times 10^{-7}$; two-sided Wilcoxon signed-rank test against zero). Brackets with asterisks show results of pairwise two-sided Mann–Whitney U tests (uncorrected) between concept neurons (CN) or visual neurons (VN) in perceptual (p) or semantic (s) trials: $p$(CNs versus CNp) $= 3.08 \times 10^{-3}$, $p$(CNs versus VNs) $= 8.97 \times 10^{-5}$, $p$(CNs versus VNp) $= 5.82 \times 10^{-4}$. ****$p < 0.0001$; ***$p < 0.001$; **$p < 0.01$. Source data are provided as a Source Data file.

presented in second position was compared to the reactivated response of the other neuron (n1) whose preferred stimulus had been shown in first position. Correlations were positive and significantly different from zero for pairs within, but not across wire bundles of the same hemisphere ($p = 0.022$ vs. $p = 0.983$; Wilcoxon signed-rank test, Fig. 5b, boxplots in dark blue). Similarly, pairwise normalized firing rates of all response-reactivation trials of these pairs were correlated strongly within ($p < 0.0001$; Pearson correlation), but not across wire bundles ($p = 0.981$; Fig. 5b, scatter plots in dark blue). Correlation strengths did not differ significantly between perceptual and semantic trials ($p = 0.089$ within, $p = 0.319$ across wire bundles; Mann–Whitney U test; Supplementary Fig. 3b). During trials in which neither preferred concept was shown, on the other hand, pairwise correlations did not exceed chance, neither within ($p = 0.168$) nor across ($p = 0.071$) wire bundles (Fig. 5b, boxplots in yellow). Remarkably, correlations within bundles were always significant if the second picture showed the preferred concept of neuron n2, even if the first picture did not depict the preferred concept of neuron n1 (light blue, $p < 0.001$ within, $p = 0.327$ across bundles). During non-preferred second picture presentations of neuron n2, no significant correlations could be detected, even for reactivation trials of neuron n1 (turquoise, $p = 0.306$ within, $p = 0.948$ across bundles). Additionally, the distribution of positional effects resembled that of relational responses across brain regions and experimental conditions (Fig. 5c). Namely, positional effects were observed in the activity of concept neurons of all brain regions except for the parahippocampal cortex during the main-comparison condition, in none of the brain regions during the no-comparison condition, and only in the hippocampus during the question-comparison condition, potentially due to the effects of working memory load and attention[11].

**Reactivations after activity silence following non-specific activation.** Finally, we asked whether and how reactivations could occur after longer periods of activity silence. In our question-comparison condition, a concept contained in the question had to be compared to that of either the first or the second picture. The response prompt at the end of each trial

revealed which comparison was to be made (Fig. 1c). If the first picture depicted the preferred concept, activity of concept neurons during the response prompt differed significantly between these two alternatives ($p < 0.05$, Fig. 6a). Following complete activity silence, concept neurons were reactivated 500 ms after presentation of the response prompt but only if it referred back to the preferred concept. Meanwhile, 104 non-visually-selective neurons not responsive to any of the four pictures used per session (binwise signed-rank test) whose mean z-value exceeded one during the first 1500 ms (baseline −400 to 100 ms) sharply increased firing ~250 ms earlier than reactivated concept neurons (Fig. 6b).

## Discussion

Our results demonstrate that concept neurons in the medial temporal lobe flexibly represent abstract relations between concepts as long as the task instruction implies that this relation is currently relevant. Relations were dynamically expressed as reactivated firing in response to non-preferred stimuli with a stereotypical, late onset in approximately half of our concept neurons whenever the task instruction prompted a comparison to the preferred concept (Fig. 2a, b). Importantly, reactivations depended on the explicit recognition and further processing of concepts. They occurred most frequently in concept neurons as opposed to merely visually selective neurons, and during questions that required semantic processing of the pictures (Fig. 4b). Furthermore, we found strong evidence that the activity of concept neurons reflects which abstract concept is currently attended to. Remarkably, concepts could be maintained in working memory without sustained activation[12]. Responses even re-emerged after temporary task irrelevance and complete activity silence when the response prompt directed the subjects' attention back to the preferred concept (Fig. 6a). Temporally ordered firing patterns of concept neurons (Fig. 3b) offer a potential mechanism for such activity-silent reactivations through synaptic modifications[18–20]. Response strength to a preferred concept in second position was correlated with reactivation strength (Fig. 5b) and both concept-specific and non-specific (Fig. 6b) stimulus responses preceded reactivations. Therefore, stimulus responses could contribute to reactivations, but with highly task-specific

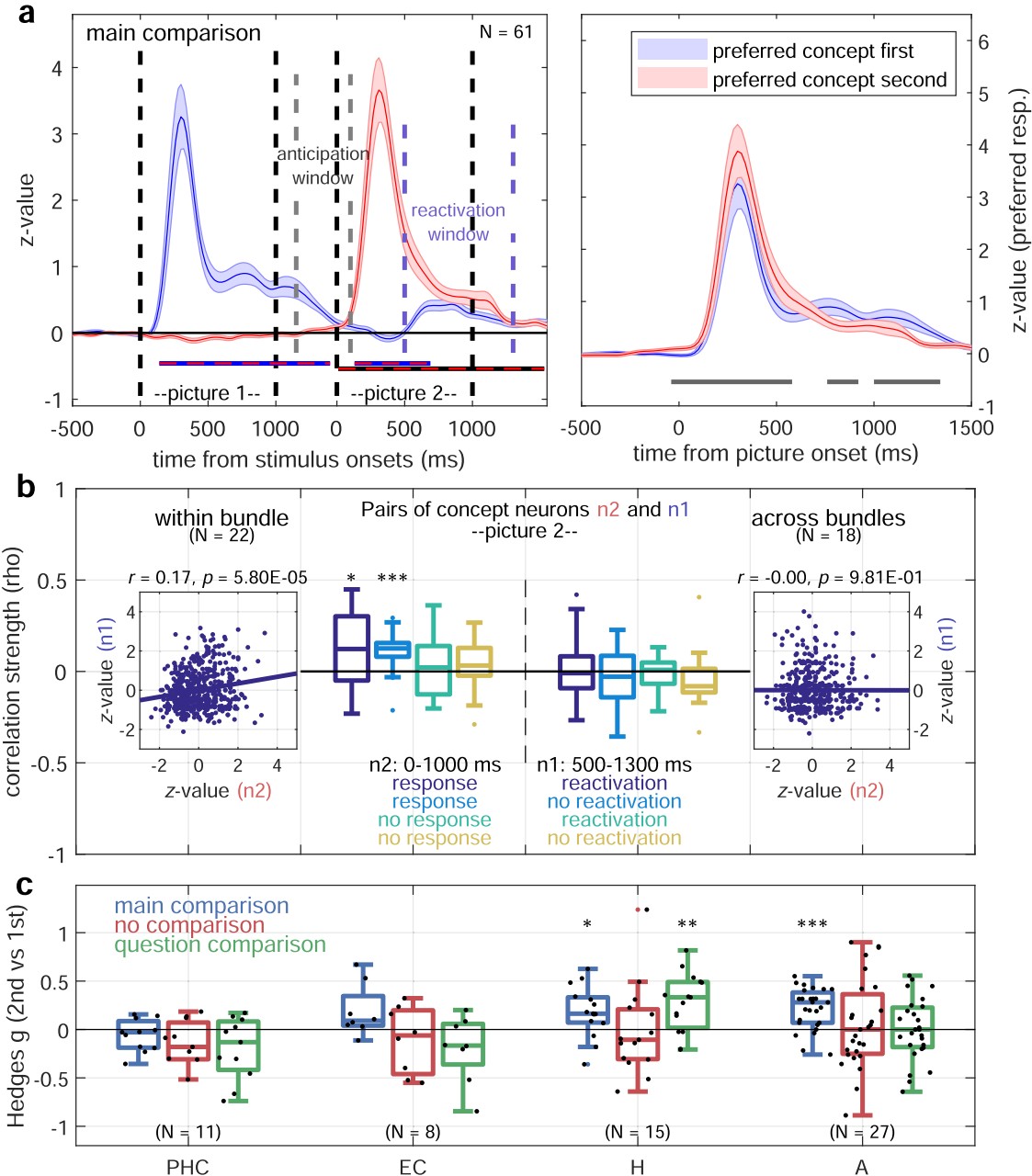

**Fig. 5 Relational responses could arise from local interactions of concept neurons. a Left:** Averaged normalized firing rates of concept neurons (mean ± SEM as shaded areas) when either the first (blue) or second (red) picture was the preferred stimulus. Time periods of significant differences are indicated by solid lines in respective colors ($p < 0.01$; two-sided cluster permutation test, black: zero). When the preferred concept was shown first, neural activity increased before the onset of the second picture (anticipation window, gray dashed lines). Second responses (red) were stronger, and their late phase coincided with reactivations of units responsive to the first picture (reactivation window, blue dashed lines). **Right:** Direct comparison of positional differences (two-sided cluster permutation test, periods of significant difference marked in dark gray). **b** Pairwise activity of concept neurons n2 and n1 with non-identical preferred stimuli during the main experiment in either the response (n2: 0–1000 ms) or the reactivation window (n1: 500–1300 ms) of picture 2. Pairs from the same hemisphere but different micro-wires were obtained from within ($N = 22$, left) or across ($N = 18$, right) wire bundles. Boxplots (Q1, median, Q3; whisker: points within ±1.5 IQR) of pairwise Spearman correlation effect sizes (rho) for different trials distinguished by color. Dark blue: both pictures preferred (n1: 1st picture, n2: 2nd picture). Light blue: only second picture preferred by neuron n2 (response). Turquoise: only first picture preferred by neuron n1 (reactivation). Yellow: none of the pictures preferred. Pairwise correlation effect sizes significantly exceeded zero within bundles whenever the second picture was preferred by neuron n2: $p(\text{dark blue}) = 2.21 \times 10^{-2}$, $p(\text{light blue}) = 7.79 \times 10^{-4}$ (two-sided Wilcoxon signed-rank tests). Inlets contain scatter plots and regression lines of normalized pairwise activity of trials when both pictures were preferred (dark blue). P values and effect size of regressions are shown on top **c** Boxplots (Q1, median, Q3; whisker: points within ±1.5 IQR) of Hedges' g of positional differences of response activity to the preferred stimulus for all brain regions and experimental conditions (two-sided Wilcoxon signed-rank test against zero). During the main condition (blue), concept neurons of all brain regions, except PHC, responded more strongly to their preferred stimulus in the second position: $p(\text{EC}) = 7.81 \times 10^{-2}$, $p(\text{H}) = 2.15 \times 10^{-2}$, $p(\text{A}) = 1.46 \times 10^{-4}$. In the no-comparison condition (red) no differences were present, and in the question-comparison condition (green) only the hippocampus exhibited higher firing during second picture presentations: $p(\text{H}) = 6.71 \times 10^{-3}$. ***$p < 0.001$; **$p < 0.01$; *$p < 0.05$ (**b**, **c**, uncorrected). Source data are provided as a Source Data file.

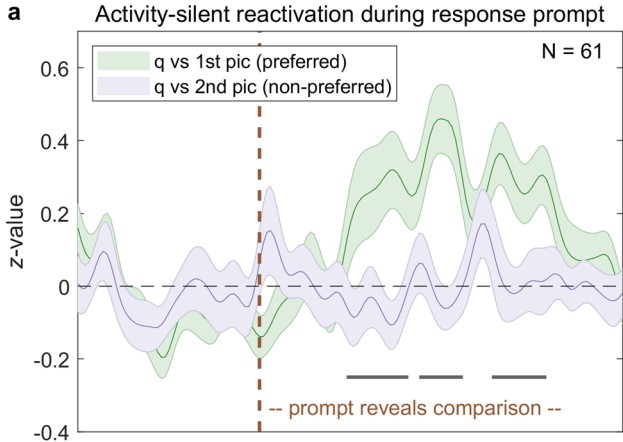

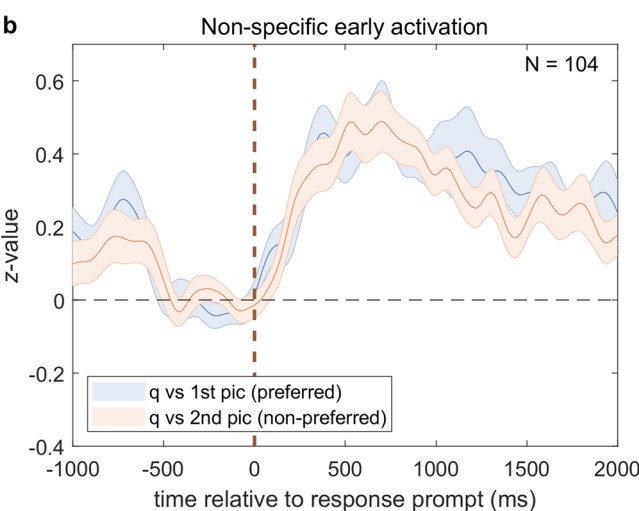

**Fig. 6 Reactivations can occur after activity silence following non-specific activation.** Concept neurons are reactivated after activity silence (**a**) following non-specific activation (**b**) whenever attention is directed back towards their preferred concept. **a** Averaged normalized firing rates of all concept neurons with standard errors (shaded areas) in the question-comparison condition during the presentation of the response prompt at the end of each trial (see Fig. 1b) whenever the first picture was preferred. The response prompt (onset: vertical dashed line in dark red) asked for a comparison between the concept of the question (non-preferred) and either that of the first (preferred, green) or that of the second picture (non-preferred, violet). Normalized activity differed significantly between these two response prompts ($p < 0.05$; two-sided cluster permutation test, black) and concept neurons were reactivated only if the prompt referred back to the preferred concept (green). Following complete activity silence, reactivations began 500 ms after presentation of the response prompt (see also reactivation window in Fig. 5a). **b** Same as **a** but with averaged normalized firing rates of non-visually-selective neurons whose $z$-value exceeded one during the response prompt (0–1500 ms; onset: vertical dashed line in dark red). Blue and red again denote comparisons to first (blue) or second (red) picture concepts. Activity sharply increased after ~250 ms, roughly 250 ms before reactivations of concept neurons, and did not differ between the two response prompts ($p < 0.05$; two-sided cluster permutation test). Data are presented as mean values ± SEM (**a**, **b**) with solid lines and shaded areas, respectively. Source data are provided as a Source Data file.

neural trajectories. Overall, our data support the notion that sequential activation of concept neurons leads to the context-dependent storage of meaningful relations between elements of experience.

**The medial temporal lobe and working memory in humans.** In our task, subjects had to maintain the identity and attributes of concepts and their relations in working memory. While working memory has traditionally been mainly attributed to the prefrontal cortex[23], more recent findings indicate it is a distributive process[12] and both electrophysiological and lesion evidence point to an involvement of the human medial temporal lobe (MTL). Lesions of the MTL mainly affect long-term memory[6], yet working memory is compromised as well under high memory load, interference by other memory items, and longer time intervals[7–9]. These diverging findings are integrated by Cowan's model of working memory, which stresses interactions between the central executive and long-term memory[24]. A recent review by Kaminski and Rutishauser summarizes evidence for this model in humans[10] and discusses putative mechanisms[20]. According to the authors, persistent activity accounts for the maintenance of memory items within the focus of attention, dynamic activity reflects attentional switches of the central executive, and synaptic-plasticity mechanisms within the MTL enable the recovery of the information that was lost from the active WM buffer. Since our task is complex with frequent changes of attentional focus and relatively high memory load, it is well suited to assess these potential contributions of the MTL to working memory.

**Persistent activity reflects attention.** Visually selective neurons in the human MTL maintain preferred visual stimuli in working memory through persistent firing[11–13]. Their maintenance activity predicts memory performance[11,12], yet persistence of firing is associated with high metabolic costs[25] and not always sustained continuously. Specifically, presentations of non-preferred stimuli interrupt this persistent activity[12], and it decreases under high memory load[11]. Multiple aspects of our data confirm the hypothesis that firing of concept neurons only persists when the preferred concept lies within the current focus of attention. First, activity was only sustained when the preferred concept needed to be processed semantically, both during the main and question control experiment (Fig. 2a). Second, when attention was diverted towards fixation crosses (no-comparison control), responses were attenuated and short, and did not entail subsequent internal reactivations (also see theories of depth of processing[26]). Third, stereotypical internal reactivations occurred as soon as comparisons to the preferred concept had to be performed (Fig. 2a, b), even retrospectively during the question control (Fig. 6a, Supplementary Fig. 5, see also the next section), and were consistent across experimental conditions (Fig. 3c). Finally, concept neurons increased firing when their preferred concept was about to become relevant. Anticipatory single-unit activity has been reported for repeated presentations of visual stimuli in fixed sequences[27]. In our paradigm, however, pairs of concepts were depicted randomly. While the identity of the second picture was constrained by that of the first, it could only be inferred stochastically (probability of 1/3 instead of 1/4). If the preferred concept was not show first, it was more likely to be shown second, and firing increased shortly beforehand (Fig. 5a). This is consistent with a pre-allocation of attention for upcoming comparisons[28] and supports the notion that the MTL is involved in inference, planning, and imagination[29,30].

**Reactivations of concept neurons after activity silence.** Bulk imaging studies of working memory suggest that reactivations of maintained memory items can occur after periods of activity silence whose duration exceeds that of non-preferred stimulus presentations[18]. It has been asked whether the same applies to the firing of concept neurons with lower detection thresholds[10]. While true for single neurons during free recall[31], our question-

comparison control sheds light onto the reactivation dynamics of concept neurons in working memory. A concept contained in the question had to be compared separately to each picture concept before a prompt revealed which of the two comparisons had to be reported. In contrast to the main condition, first picture presentations of the preferred concept were not associated with persistent firing (Fig. 2a, green versus blue) further supporting the attention hypothesis outlined above. Specifically, attention was diverted away from the preferred concept when a second non-preferred picture had to be compared to a non-preferred concept mentioned in the initial question. Remarkably, activity then re-appeared after complete activity silence upon presentation of the response prompt if and only if the first comparison (between question and first preferred picture) had to be reported (Fig. 6a). Concepts were thus maintained in working memory despite activity silence of concept neurons (see also Supplementary Fig. 6).

**Alternatives to persistent activity**. Reactivations after activity silence point to hidden memory states residing either within cells themselves[21,22,32] or within the network[20]. Assuming the latter, memory items could be maintained either outside (e.g., in inferotemporal[33], perirhinal[34,35], parietal[36], or prefrontal cortex[37]) or within the MTL in either dynamical or coherent firing, or in distributed patterns of synaptic weights[20]. For humans, visually selective neurons in the MTL may contain information about memory items via firing at preferred phases of ongoing theta oscillations under high memory load[38]. Supporting synaptic mechanisms, trial-relevant but unattended memory items can be reactivated after activity silence by a single pulse of TMS[18]. Our findings are consistent with such a state-depended network response. Reactivations were preceded either by specific activation of other concept neurons (Fig. 5) or by increased firing of non-visually-selective neurons within the MTL (Fig. 6). Non-specific activation during instruction cues has previously been reported[11] and could facilitate reactivations[18,39]. Trial-wise correlation patterns between concept neurons in our data support the idea that the presentation of non-preferred stimuli could contribute to reactivations (Fig. 5b, Supplementary Fig. 2c). Specifically, the activity of pairs of concept neurons n2 (pic2: 0–1000 ms) and n1 (pic2: 500–1300 ms) of the same hemisphere exhibited trial-wise correlations after the onset of the second picture whenever neuron n2's preferred stimulus was shown, regardless of whether the preferred stimulus of neuron n1 was shown first (reactivation) or not (Fig. 5b, dark and light blue). If neurons n2's preferred picture was not shown in second position, no pairwise correlations were present (Fig. 5b, turquoise and yellow). Correlations thus appear to depend on the preferred-stimulus responses of neuron n2, but not on the previous activation of neuron n1. Correlation patterns were thus no sufficient cause, but did plausibly contribute to reactivations, together with additional factors such as changes in excitability or intrinsic plasticity of neuron n1 due to recent activation[22,40] (see also Supplementary Fig. 4), question-specific (top-down) input to concept neurons, or the strengthening of task-specific dynamical pathways via synaptic modifications resulting from sequential activation of (concept) neurons (Fig. 3b).

**Distribution across MTL regions and potential significance of relational responses**. We found relational responses in the amygdala, entorhinal cortex and hippocampus. As suggested previously[11], the amygdala could represent concepts (and their relation) in working memory. On the other hand, while responses to concepts were selective and invariant, responding to both pictures and written names and thus likely multimodal[41], it is unclear whether

information about concepts per se is encoded. Single neuron activity in amygdala also correlates with subjective value[42] in humans. If a subset of concept-specific amygdala neuron responses encoded an emotional dimension rather than concept identity, this information could be imparted onto specific concepts via relational responses. The entorhinal cortex, on the other hand, partakes in the formation of associative memories[43]. It represents a highly processed version of the ongoing sensory experience to hippocampus[44]. The hippocampus itself is important for the encoding[45], retrieval[46,47], and consolidation[48] of episodic memories. In our study, relational reactivations were most frequent among concept neurons in the entorhinal cortex and hippocampus hinting at their potential contributions to both activity-silent retrieval of concepts in working memory and episodic memory.

**Higher-order data structures: potential as a network of concept tags**. Relational processing did not appear to permanently alter the tuning and selectivity of concept neurons, contrasting a recent report on visually selective neurons during associative learning in humans[14]. While our experiment imposed meaningful relations between concepts, they were frequently altered from trial to trial and only needed to be remembered for a short period of time. However, input-driven early-onset responses and dynamic late-onset reactivations resulted in reliable sequential (reverse) firing of pairs of concept neurons on a trial-by-trial basis (Fig. 3b). This way, synaptic modifications via spike-timing or behavioral timescale-dependent plasticity could lead to the storage of meaningful conceptual relations. Concept neuron activity reflected both current and past stimuli in hippocampus, a prime candidate for hosting contextual pointers. Both visually selective and non-visually selective neurons in humans recapitulate activity states during encoding[49], and single neuron activity of the hippocampus leads to cortical reinstatement in entorhinal cortex[15]. Moreover, drift and reinstatement of contextual representations in the human MTL have been linked to memory performance[18,50]. Recapitulating a past neural pattern is thought to increase the accessibility of preceding and successive memories due to its high similarity to preceding or successive neural patterns[51]. Similarly, remembering a word from a studied word list increases the likelihood of remembering the preceding and following words as a function of their temporal distance to the recalled word[52]. Rodent research identified two hippocampal cell types that could organize contextual representations according to time and space, namely time and place cells. The question arises how reference of memories by time and space alone could account for the flexible retrieval of memories in completely different spatiotemporal contexts. Semantic[53] or even subjective associations[54] between words influence the recall performance of studied word lists. Remembering a concept from a studied word list increases the likelihood of recalling semantically or subjectively similar concepts from the same list (or even from different lists)[55]. Similarly, responses of visually selective neurons to multiple pictures in humans often occur when subjects report a semantic or subjective connection between these pictures that is stronger than for other pictures[56,57]. Given the highly abstract nature of relational responses in this study, concept neurons could unite spatiotemporally different episodes through their conceptual connection. This way they could partake in the formation of the intricate narratives of our lives in which episodic memories are highly interconnected and cross-referenced.

## Methods

**Subjects and recording**. All studies were approved by the Medical Institutional Review Board at the University of Bonn (accession number 095/10 for single-unit recordings in humans in general and 248/11 for the current paradigm). Each patient gave informed written consent both for the implantation of micro-wires

and for participating in the experiment. We recorded from 12 patients with pharmacologically intractable epilepsy (11 right handed, 1 ambidextrous; 6 male; 22–65 years old), implanted with intracranial electrodes to localize the seizure onset zone for surgical resection)[58]. Each depth electrode contained a micro-wire bundle consisting of nine micro-wires, including a reference with low impedance and eight high-impedance recording electrodes (AdTech, Racine, WI), which protruded from the tip of the electrode by ~4 mm. All bundles were localized using a post-implantation CT scan co-registered with a pre-implantation MRI scan normalized to Montreal Neurological Institute space. The differential signal from the micro-wires was amplified using a Neuralynx ATLAS system (Bozeman, MT), filtered between 0.1 Hz and 9,000 Hz, and sampled at 32 kHz. These recordings were stored digitally for further analysis. Our data set consisted of 38 experimental sessions with recordings from the amygdala (A), parahippocampal cortex (PHC), entorhinal cortex (EC), and hippocampus (H). We used the spike-sorting software Combinato[59] with default parameters for the exclusion of noisy recording channels, artifact removal, spike detection, and spike sorting. Afterward, we manually removed remaining artifacts, merged potentially over-clustered units from the same channel and distinguished single from multi-units using Combinato's graphical user interface. Highly similar spike shapes, inter-spike interval distributions, neural responses to visual stimuli, asymmetric cross correlations as well as the absence of neural activity during refractory periods guided this procedure which predated all further analyses.

**Experimental design**. The paradigm consisted of a main part and two control conditions performed on a laptop computer. We used Psychtoolbox3 (www.psytoolbox.org) and Octave (www.gnu.org/octave) running on a Debian 8 operating system (www.debian.org) for stimulus delivery. Prior to the experiment, ~100 pictures of persons, animals, scenes, and objects were presented on a laptop screen in pseudo-random order (presentation for 1 s; 6 or 10 trials). Then, after automatic spike extraction and sorting with Combinato, neural responses to these pictures were evaluated based on raster plots and histograms. The aim of this procedure was to identify a subset of four pictures for the following experiment while maximizing the number of neurons that were expected to respond selectively to only one of the pictures.

**Main-comparison condition**. Each self-triggered trial contained one out of five questions, a sequence of two of four pictures with jittered onsets and an answer prompt displaying "1 or 2?". Subjects indicated the sequential position of the picture that best answered the question by pressing key 1 or 2. We resolved ambiguous meanings of each depicted picture before the experiment and elaborated on the meaning of each question in a short test run of the paradigm. The questions were "Bigger?" (volume), "Last seen in real life?", "More expensive?" or "Older?" (if the pictures set included a person), "Like better?", and "Brighter?". Subjects were instructed to try to stick to one answer for a given picture pair, but to keep mentally computing the answer to each question in the course of the experiment. In total, the experiment consisted of 300 trials in which each of the five questions and all 12 possible ordered picture pairs out of four pictures were presented equally often and in an unpredictable pseudo-random order. This resulted in 60 trials per question, 25 trials per picture pair and 5 trials per specific combination of question and picture pair.

**No-comparison control condition**. In 60 additional trials, subjects were instructed to count the number of red fixation crosses. None, one, or both of them could be red instead of white. Depending on the number of red fixation crosses, the answer prompt showed "1: one 2: none" or "1: one 2: two" with equal probability and such that each answer was equally likely to be the correct one.

**Question-comparison control condition**. As with the no-comparison control condition, only questions and answer prompts differed from the main condition. Each question contained one of three question elements, namely "Bigger?", "More expensive?" or "Older?" and "Like better?", combined with a word referring to one of the pictures according to the patients' judgment. Two examples of the resulting structure of the question are: "Chair is bigger than item in…" or "Keyboard is more expensive than item in…". The response prompt following the two picture presentations referred to either "… the first picture" or "the second picture" and asked for a key press with the instruction "1: yes 2: no". Each of the 36 combinations of question and picture-pair were presented twice in pseudorandom order, resulting in 72 trials.

**Analysis of neural data**. Visually selective neurons were defined by two criteria. First, a bin-wise signed-rank test detected neural picture responses against baseline (−400 to 100 ms) in the main condition, emphasizing the reproducibility of their onset and time course over trials. Specifically, nineteen overlapping 100-ms-bins in a response window ranging from 0 to 1000 ms post stimulus were compared to baseline[60] in a Wilcoxon signed-rank test (alpha = $10^{-5}$, Simes corrected, "signrank" in MATLAB 2016b). Thus detected neurons had to respond to exactly one of the pictures. Second, firing rates during the presentation of this preferred picture had to exceed those of all remaining pictures (Hedges' $g > 0.3$), both in the main condition and during controls. Hedges' $g$ describes the difference of means relative to the pooled variance. Concept neurons were defined as visually selective neurons

with higher firing during questions of the question-comparison control that included the written name that referred to the preferred concept versus all other written names (Hedges' $g > 0.3$). Visually selective neurons that were not concept neurons were termed visual neurons. Concept neurons with higher firing 500–1300 ms after second picture presentations in relational (first picture = preferred stimulus) versus non-relational (preferred picture not shown) trials (Mann–Whitney U test, alpha = 0.05, "ranksum" in MATLAB 2016b) were termed reactivated neurons. Concept neurons that were no reactivated neurons were termed input-driven neurons. Non-visually-selective neurons as defined in Fig. 6 did not respond to any picture (bin-wise signed-rank test) but instead exhibited normalized firing relative to baseline (−400 to 100 ms) exceeding one during the 1500 ms upon presentation of the response prompt.

**Relational responses at the population level**. Binned peri-stimulus firing rates (1 ms bins) were normalized relative to the baseline interval before first picture presentations (−500 to 0 ms). Resulting $z$-values of concept neurons were averaged across relational trials (preferred stimulus shown first) and convolved with a Gaussian kernel (sigma = 50 ms). Afterward, pairwise $z$-score differences between experimental conditions were evaluated with a cluster permutation test[61]. Paired samples $t$-tests (two-sided) were computed for each time bin (alpha = 0.01) to determine clusters of contiguous $z$-score time bin differences. Sums of $t$-values within each cluster were assessed relative to the null distribution of $t$-value sums obtained from 1000 permutations of experiment labels. If $t$-value sums fell into the top percentile of the permutation distribution, $z$-value differences during the time period of the respective cluster were considered significant. Responses (0–1000 ms after stimulus onset) and reactivations (500–1300 ms after stimulus onset) obtained from convolved normalized activity were compared during different stimulus presentations and experimental conditions (Fig. 2c). Pairwise relationships were depicted as scatter plots and quantified via Pearson correlation strengths visualized as regression lines.

**Cross-correlations**. Cross-correlations were computed on a trial-by-trial basis for pairs of concept neurons from different micro-wires of the same micro bundle (0–1500 ms after picture onsets). Only trials in which both preferred concepts of each neuron appeared were analyzed, this was done separately for both picture orders. Concept neuron pairs with identical preferred stimuli were excluded. The number of coincident spikes in millisecond time bins (multiplied by 1000) for lags up to 750 ms were normalized to the mean firing rates of both neurons (baseline interval of −500 to 0 ms from the main condition). Each cross-correlogram was then convolved with a Gaussian kernel (sigma = 10 ms) and averaged over trials. Differences of cross-correlations between experimental conditions were again determined by the cluster-based permutation test described above. Cross-correlations of non-local pairs of concept neurons across wire bundles were computed analogously (Supplementary Fig. 2a). In order to capture stimulus-independent interactions, shift predictors were obtained by calculating cross-correlograms from non-simultaneous pairwise activity of consecutive trials[62]. After baseline normalization (geometric mean) and convolution (100 ms boxcar kernel) they were subtracted from simultaneous cross-correlograms for non-local (Supplementary Fig. 2b) and local (Supplementary Fig. 2c) pairs of concept neurons and compared between experimental conditions (cluster permutation test, alpha = 0.05).

**Statistics and reproducibility**. All 61 detected concept neurons were distributed across 26 sessions of 10 patients. Eleven sessions contained at least two pictures that evoked unique responses (Supplementary Fig. 1).

**Reporting summary**. Further information on research design is available in the Nature Research Reporting Summary linked to this article.

## Data availability

All data supporting the findings of this study are publicly available at https://github.com/mabausch/ConceptNeuronRelations.git. Source data are provided with this paper.

## Code availability

All code related to the analyses of the manuscript is available at https://github.com/mabausch/ConceptNeuronRelations.git.

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

## Acknowledgements

We thank all patients for their participation. This research was supported by the Volkswagen Foundation, the German Ministry of Education and Research (BMBF 031L0197B) and the German Research Council (DFG MO 930/4-2, SPP 2205, SFB 1089).

## Author contributions

M.B., J.N. and F.M. designed research. C.E.E. and F.M. recruited and managed patients. J.B. and F.M. performed surgeries. M.B., T.P.R., S.M. and J.N. collected data. M.B. analyzed data. F.M. supervised project. M.B. and F.M. wrote the manuscript.

## Funding

## Competing interests

The authors declare no competing interests.
