## [Peer Review File · Nature Communications]

Concept neurons in the human medial temporal lobe flexibly represent abstract relations between conceptsREVIEWER COMMENTS

Reviewer #1 (Remarks to the Author):

The paper by Bausch et al. presents data from an impressive set of recordings in 12 patients – 2512 neurons in all. They concentrated their analysis on 61 neurons “concept” neurons, responding selectively to the presentation of specific visual images. The key finding was that when participants were requested to make a comparison between two images (out of a set of four), there was a response to the presentation of the non-preferred image, but with a substantial delay of around 400 ms compared to the standard response.

The paper is well-written, the figures are good, and overall, I was impressed by the work.

The results are certainly intriguing and would appear to the existing literature on concept neurons. We already knew that mentally reactivating the memory of a particular stimulus activates a selective neuron, so that, in a sense, this result is not a total surprise. However, what is truly novel is the design that allows the time course of the response to be examined in detail.

This sort of data is very precious, and so I am favorable to the idea of publishing it in Nature Communications.

However, I do have some questions.

Firstly, I found the authors somewhat unclear about how they interpret the activation. Does the activation correspond to the internal reactivation of the concept? If this is what the authors believe, then it might be worthwhile making this explicit in the text.

A second question concerns how the experiments are run. If I understood correctly, each testing session uses a fixed set of 4 images, chosen from an initial collection of around 100 images because they were images that produced selective visual responses in the recorded neurons. However, it is unclear how many of the images resulted in a selective response. I presume that, ideally, there would be a least four different neurons, with one at least responding selectively to each of the four stimuli. But, this may not be the case. The protocol could presumably be done even with only one responsive neuron.

This may not be a problem, but looking at the data, it seems that this could be critical. In the data presented in figure 1, it is clear that there are a huge number of repeats with one particular stimulus – the “tie”. And there are even a large number of trials where a strawberry is presented after the question “Bigger” followed by “Tie”. If these trials were mixed together with trials designed to ask the other questions “Like Better”, “Last seen in real life?” etc., then the number of trials required would become very large indeed.

I have the suspicion that quite a bit of the data may have been obtained under conditions where, the vast majority of trials will have been identical – simply to get enough data to plot the results. But if the question and the stimuli on any trial (Bigger followed by Tie and Strawberry) are nearly always the same, then the protocol would become highly predictable. There are already studies showing that such neurons can effectively respond in anticipation of upcoming events, so doesn't this mean that such phenomena could be occurring here?

I'm not sure that this would be enough to invalidate the findings. After all, the population-based responses illustrated in figure 2A do seem to show a reliable difference in the response to the second stimulus depending on whether there is a comparison to be made.

If the authors can reply to these points, I feel that the paper could be suitable for publication.

Reviewer #2 (Remarks to the Author):

Bausch et al conducted single neuron recordings in awake behaving epilepsy patients implanted to localize their seizure foci. They examined the mechanisms of semantic linking between ‘concepts’ in a task dependent manner. They find that subsets of ‘concept cells’ across the medial temporal lobe (in the Hippocampus, Amygdala, Entorhinal, and Parahippocampal cortex) show a novel and not before reported kind of response to non-preferred stimuli when that stimulus is related to the preferred stimulus through a task. They show data to argue that this is a reflection of relational processing in a demand-dependent manner.

Main results are: (1) During the main comparison task condition, concept cells develop a delayed response to the non-preferred stimulus when the preferred stimulus is presented first. The authors call this a ‘reactivation’ response. It can be seen both at the single unit and population level (Figs 1 and 2 respectively). (2) This can be observed on a trial by trial basis both amongst pairs of local concept neurons and at the local population level. Correlated activity of pairs of concept neurons was only seen

when the pairs were local (from the same microwire bundle) (Figure 3 and 5b). (3) Comparisons need to be 'semantic' instead of 'perceptual' for concept cells to show relational responses (Fig 4). (4) Most cells had higher responses when their preferred object was in the second position. The strength of the second response was related trial-by-trial to the strength of the reactivation response of pairs of neurons within same but not different bundles, arguing for a kind of 'all or nothing' reinstatement.

Overall assessment: Overall, I find the task to be very elegant and the overall findings quite compelling. The findings are novel and of broad interest. The recordings are of high quality and the number of cells included is impressive (and needed, given that 33/2512 cells showed the effect reported). Some technical and writing issues need to be addressed (in particular re Fig 3 and 5, analysis of which is not worked out in detail). I am supportive of publication after these issues are addressed.

Major concerns:

1. Framing of results. The introduction and discussion makes reference to concepts that are not of relevance and not tested (i.e. optogenetics, episodic, synaptic). In particular, there is extensive discussion of episodic memory, long-term memory, engrams of such etc. But it is not clear to me what the work presented has to do with episodic memory – the task here is a working memory task and has neither a long-term nor episodic component. It would seem more prudent to anchor these findings in the context of working memory literature, which is only mentioned in passing. Reactivating representations of items currently being manipulated in WM such as here is clearly highly interesting (and the mechanisms of such are hotly debated, as is the role of the MTL in WM in general). It isn't clear to me why the results are not put into this proper context in discussion/intro, which instead extensively discusses episodic memory and synaptic mechanisms.

2. Population response (Fig. 2). I have a few questions of clarification for this figure. Fig 2A: is this for all $n=61$ concept cells or only the 33 that showed the response? 2B: here it is clear that all $n=61$ are shown. I find it hard to see how robust the effect is across cells in Fig 2. It would be better to show this as a scatter to be able to appreciate the full distribution of the kinds of responses visible (showing reactivation). Also, was the 1/3rd of concept cells that showed responses to non-preferred images when related to the question the same than those that showed reactivation to pictures? (comparing 2A+B; green line in 2A shows no response?).

3. Cross-correlation analysis and claims that such interactions are local (Fig. 3). First, the claim made is that this can only be seen for local pairs. But this is not substantiated quantitatively. What does 3B look like for non-local pairs? More broadly, why would one only expect this to occur on local pairs? Second, it wasn't clear to me whether these cross-correlograms were corrected for what would be expected by covariation of firing rates by subtracting the shift predictor (for example, see Hirabayashi & Miyashita,

2005). If the shift predictor takes the same shape the interpretation of the data would rather be that the cells co-vary in firing rate due to some form of common input.

4. Comparison of concept cell reactivation responses between brain areas. First, Fig 4A is expressed in terms of # units. But the total # units recorded in each area presumably differs. To substantiate this claim, would have to show that this also holds as a proportion of recorded neurons. Second, some more details need to be given for this analysis. For Fig 5B, how many neurons are included (i.e. only the 33 reactivated or all 61)? Also is what is plotted only for the preferred stimulus? If so, is there an effect on the non-preferred responses?

5. Claim of local interactions (Fig 5B). First, the finding of enhanced responses to preferred stimuli if they were shown in the second position only during the main condition (Fig 5A,C) is very interesting. Was this seen during both the semantic and the perceptual comparison questions or was this also related to the semantic question only like in Fig 4B? Second, Fig 5B makes the claim that the extent of the reactivation response to non-preferred stimuli is positively correlated with the response in the same trial of concept cells that prefer the second stimulus. This striking phenomena deserves more detailed analysis (scatter, or PSTH). Is there temporal overlap between the two responses or is there a 'silent' delay in between? If there is no delay I cannot see how the claim of 'activity-silent synaptic mechanisms' (discussion) can be supported. A more straightforward explanation would be that one neuron excites the other, and that this excitation is somehow gated by this neuron having been active shortly before (first stimulus)? Perhaps this is what is meant, but this should be described more carefully. Also, the claim is advanced that 'reactivation' only happens during the semantic trials of the main condition. How is this taken into account for Fig 5B, i.e. since there is no reactivation during perceptual, there should also be no such correlation in the local condition?

Minor issues:

1. Definition of 'concept neurons'. The definition of a concept neuron here is one that responds vs. baseline selectively to only a given picture as well as also to instructions (text) that feature that object. It would be useful to compare this definition with others used in the literature so far (i.e. multimodal? Different images to same object).

2. concept cell firing to different photographs involving the concept 'airplane'.

3. In the introduction the authors seem to be downplaying the current state of the field with regard to relational processing in the hippocampus. While such task dependent relational firing has not been demonstrated prior, statements like "It is currently not known however, whether or how concept neurons dynamically encode conceptual relations and could thereby account for the integrative properties of episodic memories" ignores the fact that it is known that hippocampal concept cells fire to related concepts. Papers like Ison et al 2015, Staresina et al 2019 (both of which the authors cite later), and Rey et al 2018 are good examples of human single unit papers that demonstrate this.

4. For many figures it is not clear to me what the n is. This should be stated or individual datapoints shown (i.e. Figure 4b, 5b, 5c).

5. Line 167 – conceptual processing needs to be defined – how is this different from semantic processing?

6. Line 189-190 – “On the other hand, relational signals likely enter the hippocampus when episodic memories are formed” This needs a citation.

7. Line 221-223 – This sentence “...neural drift” needs a citation.

REVIEWER COMMENTS

Reviewer #1 (Remarks to the Author):

The paper by Bausch et al. presents data from an impressive set of recordings in 12 patients – 2512 neurons in all. They concentrated their analysis on 61 neurons “concept” neurons, responding selectively to the presentation of specific visual images. The key finding was that when participants were requested to make a comparison between two images (out of a set of four), there was a response to the presentation of the non-preferred image, but with a substantial delay of around 400 ms compared to the standard response. The paper is well-written, the figures are good, and overall, I was impressed by the work. The results are certainly intriguing and would appear to the existing literature on concept neurons. We already knew that mentally reactivating the memory of a particular stimulus activates a selective neuron, so that, in a sense, this result is not a total surprise. However, what is truly novel is the design that allows the time course of the response to be examined in detail. This sort of data is very precious, and so I am favorable to the idea of publishing it in Nature Communications.

We thank the reviewer for the succinct summary and positive assessment of our manuscript. Based on the insightful comments and questions regarding design features and unit statistics, we extended the manuscript and figures. Specifically, the questions addressing the interpretation and predictability of activations have led to a new finding (anticipatory activity) and have improved the clarity of our manuscript, for which we are grateful. Furthermore, we now address additional aspects of concept neuron activity such as factors contributing to reactivations, the influence of attention and the existence of reactivations after periods of activity silence following non-specific activation.

1. However, I do have some questions. Firstly, I found the authors somewhat unclear about how they interpret the activation. Does the activation correspond to the internal reactivation of the concept? If this is what the authors believe, then it might be worthwhile making this explicit in the text.

Since reactivations only occur whenever a comparison to the preferred concept is instructed by the task (as opposed to a comparison to perceptual features of the preferred stimulus), we believe our data is in accordance with an internal reactivation of the concept and now state this more clearly in the discussion section:

“Discussion [...]

Persistent activity reflects attention [...]

Multiple aspects of our data confirm the hypothesis that firing of concept neurons only persists when the preferred concept lies within the current focus of attention. First, activity was only sustained when the preferred concept needed to be processed semantically, both during the main and question control experiment (Fig. 2a). Second, when attention was diverted towards fixation crosses (red cross control), responses were attenuated and short, and did not entail subsequent internal reactivations (also see theories of depth of processing²⁶). Third, stereotypical internal reactivations occurred as soon as comparisons to the preferred concept had to be performed (Fig. 2a-b), even retrospectively during the question control (Fig. 6a, Supplementary Fig. 5, see also the next section), and were consistent across experimental conditions (Fig. 3c).”

2. A second question concerns how the experiments are run. If I understood correctly, each testing session uses a fixed set of 4 images, chosen from an initial collection of around 100 images because they were images that produced selective visual responses in the recorded neurons. However, it is unclear how many of the images resulted in a selective response. I presume that, ideally, there would be a least four different neurons, with one at least responding selectively to each of the four stimuli. But, this may not be the case. The protocol could presumably be done even with only one responsive neuron.

We thank the reviewer for this important point. The distribution of the number of response-eliciting pictures (among the 4 chosen pictures) over sessions is now depicted in Supplementary Figure 1a. The distribution of concept neuron responses themselves can now be appreciated in Supplementary Figure 1b. At least one concept-neuron response (to one of the four depicted concepts) was detected in 26 sessions, at least two in 15 sessions (Supplementary Figure 1b). Moreover, at least two pictures evoked unique concept neuron responses in 11 sessions (Supplementary Figure 1a). It is worth noting that pairs of concept neurons with different preferred stimuli could be analyzed in two opposite orders for interaction analyses. The distribution of all ordered pairs of concept neurons over sessions is depicted in Supplementary Figure 1c. We now describe these response statistics in the results and methods section:

“Results [...]

Concept neurons indicate the presence of a relation to their preferred concept

Overall we identified 128 visually selective units. Of these, 61 units qualified as concept neurons (Supplementary Fig. 1). Thirty-three of these concept neurons responded to non-preferred concepts with a delayed but well-defined onset whenever the task required a comparison to the response-eliciting concept (approximately 400 ms later than the response to the preferred stimulus; Fig. 1d).”

“Methods [...]

Analysis of neural data

Definition of unit types

[...] Concept neurons were defined as visually selective neurons with higher firing during questions of the “question comparison” control that included the written name that referred to the preferred concept versus all other written names (Hedges’ $g > 0.3$). **All 61 detected concept neurons were distributed across 26 sessions, 11 of which contained at least two pictures that evoked unique responses (Supplementary Fig. 1).** Visually selective neurons that were not concept neurons were termed “visual neurons”.”

Supplementary Figure 1. **a.** Distribution of the number of response-eliciting pictures with at least one unique concept-neuron response across sessions. In 11 sessions, 2 or more pictures elicited at least one unique concept-neuron response. **b.** Distribution of concept-neuron responses across sessions (including cases where multiple concept neurons respond to the same picture). In 15 sessions there were 2 or more concept neuron responses. **c.** Distribution of ordered concept-neuron pairs with different preferred stimuli.

This may not be a problem, but looking at the data, it seems that this could be critical. In the data presented in figure 1, it is clear that there are a huge number of repeats with one particular stimulus – the “tie”. And there are even a large number of trials where a strawberry is presented after the question “Bigger” followed by “Tie”. If these trials were mixed together with trials designed to ask the other questions “Like Better”, “Last seen in real life?” etc., then the number of trials required would become very large indeed. I have the suspicion that quite a bit of the data may have been obtained under conditions where, the vast majority of trials will have been identical – simply to get enough data to plot the results. But if the question and the stimuli on any trial (Bigger followed by Tie and Strawberry) are nearly always the same, then the protocol would become highly predictable.

The predictability was minimized by the following design features that ensured only 5 repetitions of each combination of question and picture pair: All four stimuli and all questions were presented equally often in every session, in pseudo-random order, irrespective of neural response patterns. The experiment was comprised of 300 trials, 60 trials for each question. All 12 ordered pairs of the 4 pictures were combined equally often with each of the 5 questions. This resulted in 25 trials per picture pair and 5 trials per question-picture pair. For example, the question “Bigger?” was presented 5 times in conjunction with “Strawberry” followed by “Tie”. These 5 trials were scattered among 295 different trials in pseudo-random order to maximize uncertainty. Moreover, the switch cost associated with constantly changing task demands of different questions and picture pairs further decreased stereotypical processing of specific question picture-pairs. We now describe these design features more explicitly in the methods section:

“Methods [...]

‘Main comparison’ condition

[...]

During a total of 300 trials, each of the 60 possible combinations of questions and picture pairs were shown five times in pseudo-random order. In total, the experiment consisted of 300 trials in which each of the five questions and all 12 possible ordered picture pairs out of four pictures were presented equally often and in an unpredictable pseudo-random order. This resulted in 60 trials per question, 25 trials per picture pair and 5 trials per specific combination of question and picture pair.”

There are already studies showing that such neurons can effectively respond in anticipation of upcoming events, so doesn’t this mean that such phenomena could be occurring here? I’m not sure that this would be enough to invalidate the findings. After all, the population-based responses illustrated in figure 2A do seem to show a reliable difference in the response to the second stimulus depending on whether there is a comparison to be made.

We thank the reviewer for this insightful question. As the reviewer rightly suggests, reactivations only occurred if the task instruction required a comparison to the response-eliciting concept and during the presentation of various non-preferred stimuli (Fig. 2a+b). We now discuss more explicitly that concept neurons increase firing whenever the preferred concept lies within the current focus of attention under various conditions (Fig. 2a-c, Fig.5a, Fig. 6, Supplementary Figure 5). While anticipatory activity alone could not explain reactivation patterns, the reviewer raises a very important point. Even though the next question and following picture was never predictable, the identity of the second picture was stochastically constrained by what was shown first because the same concept was never shown twice in the same trial. Therefore, the first picture was chosen from 4 and the second only from 3 possible pictures. In order to quantify potential anticipatory activity before the onset of the second picture, we extended our statistical comparison of preferred concept presentations in first or second positions beyond picture presentation times in Fig. 5a. Importantly,

firing exceeds zero earlier in second versus first preferred concept positions (left panel) and a direct comparison reveals significant differences preceding the onset of the second preferred picture (right panel). We added a sentence to the results section and discuss these new findings:

“**Results** [...]”

Moreover, concept neurons responded more strongly to their preferred concept in second versus first picture positions, particularly during the early response phase immediately preceding the period of relational responses (“reactivation window”; Fig. 5a, $p < 0.01$; cluster permutation test) **and even before the onset of the second picture (“anticipation window”;** Fig. 5a).”

“**Discussion** [...]”

Persistent activity reflects attention

[...] Finally, concept neurons increased firing when their preferred concept was about to become relevant. Anticipatory single-unit activity has been reported for repeated presentations of visual stimuli in fixed sequences²⁷. In our paradigm, however, pairs of concepts were depicted randomly. While the identity of the second picture was constrained by that of the first, it could only be inferred stochastically (probability of 1/3 instead of 1/4). If the preferred concept was not shown first, it was more likely to be shown second, and firing increased shortly beforehand (Fig. 5a). This is consistent with a pre-allocation of attention for upcoming comparisons²⁸ and supports the notion that the MTL is involved in inference, planning, and imagination^{29,30}.

Fig. 5 Relational responses could arise from local interactions of concept neurons. **a. Left:** Averaged normalized firing rates of concept neurons with standard errors (shaded areas) when either the first (blue) or second (red) picture was the preferred stimulus. Time periods of significant differences are indicated by solid lines in respective colors (cluster permutation test, black: zero). When the preferred concept was shown first, neural activity increased before the onset of the second picture (“anticipation window”, gray dashed lines). Second responses (red) were stronger, and their late phase coincided with reactivations of units responsive to the first picture (“reactivation window”, blue dashed lines). **Right:** Direct comparison of positional differences (cluster permutation test, periods of significant difference marked in dark gray). [...]

If the authors can reply to these points, I feel that the paper could be suitable for publication.

Reviewer #2 (Remarks to the Author):

Bausch et al conducted single neuron recordings in awake behaving epilepsy patients implanted to localize their seizure foci. They examined the mechanisms of semantic linking between ‘concepts’ in a task dependent manner. They find that subsets of ‘concept cells’ across the medial temporal lobe (in the Hippocampus, Amygdala, Entorhinal, and Parahippocampal cortex) show a novel and not before reported kind of response to non-preferred stimuli when that stimulus is related to the preferred stimulus through a task. They show data to argue that this is a reflection of relational processing in a demand-dependent manner.

Main results are: (1) During the main comparison task condition, concept cells develop a delayed response to the non-preferred stimulus when the preferred stimulus is presented first. The authors call this a ‘reactivation’ response. It can be seen both at the single unit and population level (Figs 1 and 2 respectively). (2) This can be observed on a trial by trial basis both amongst pairs of local concept neurons and at the local population level. Correlated activity of pairs of concept neurons was only seen when the pairs were local (from the same microwire bundle) (Figure 3 and 5b). (3) Comparisons need to be ‘semantic’ instead of ‘perceptual’ for concept cells to show relational responses (Fig 4). (4) Most cells had higher responses when their preferred object was in the second position. The strength of the second response was related trial-by-trial to the strength of the reactivation response of pairs of neurons within same but not different bundles, arguing for a kind of ‘all or nothing’ reinstatement.

Overall assessment: Overall, I find the task to be very elegant and the overall findings quite compelling. The findings are novel and of broad interest. The recordings are of high quality and the number of cells included is impressive (and needed, given that 33/2512 cells showed the effect reported). Some technical and writing issues need to be addressed (in particular re Fig 3 and 5, analysis of which is not worked out in detail). I am supportive of publication after these issues are addressed.

We thank the reviewer for the positive assessment of our study. A thorough examination of our work and the highly relevant questions helped uncover important nuances to our findings, for which we wish to express gratitude.

Major concerns:

1. Framing of results. The introduction and discussion makes reference to concepts that are not of relevance and not tested (i.e. optogenetics, episodic, synaptic). In particular, there is extensive discussion of episodic memory, long-term memory, engrams of such etc. But it is not clear to me what the work presented has to do with episodic memory – the task here is a working memory task and has neither a long-term nor episodic component. It would seem more prudent to anchor these findings in the context of working memory literature, which is only mentioned in passing. Reactivating representations of items currently being manipulated in WM such as here is clearly highly interesting (and the mechanisms of such are hotly debated, as is the role of the MTL in WM in general). It isn’t clear to me why the results are not put into this proper context in discussion/intro, which instead extensively discusses episodic memory and synaptic mechanisms.

We agree that our data is of particular relevance to questions of working memory. Therefore, the main focus of the introduction and discussion has now been shifted. Whenever we still refer to episodic memory and synaptic mechanisms, the relationship to our data is now clarified, and they are anchored in discussions within the working memory literature. Particularly, the following new sections now address working memory in humans:

“Discussion [...]

The medial temporal lobe and working memory in humans

In our task, subjects had to maintain the identity and attributes of concepts and their relations in working memory. While working memory has traditionally been mainly attributed to the prefrontal cortex²³, more recent findings indicate it is a distributive process¹², and both electrophysiological and lesion evidence point to an involvement of the human medial temporal lobe (MTL). Lesions of the MTL mainly affect long term memory⁶, yet working memory is compromised as well under high memory load, interference by other memory items, and longer time intervals⁷⁻⁹. These diverging findings are integrated by Cowan’s model of working memory, which stresses interactions between the central executive and long-term memory²⁴. A recent review by Kaminski and Rutishauser summarizes evidence for this model in humans¹⁰ and discusses putative mechanisms²⁰. According to the authors, persistent activity accounts for the maintenance of memory items within the focus of attention, dynamic activity reflects attentional switches of the central executive, and synaptic-plasticity mechanisms within the MTL enable the recovery of the information that was lost from the active WM buffer. Since our task is complex with frequent changes of attentional focus and relatively high memory load, it is well suited to assess these potential contributions of the MTL to working memory.

Persistent activity reflects attention

Visually selective neurons in the human MTL maintain preferred visual stimuli in working memory through persistent firing¹¹⁻¹³. Their maintenance activity predicts memory performance^{11,12}, yet persistence of firing is associated with high metabolic costs²⁵ and not always sustained continuously. Specifically, presentations of non-preferred stimuli interrupt this persistent activity¹², and it decreases under high memory load¹¹. Multiple aspects of our data confirm the hypothesis that firing of concept neurons only persists when the preferred concept lies within the current focus of attention. First, activity was only sustained when the preferred concept needed to be processed semantically, both during the main and question control experiment (Fig. 2a). Second, when attention was diverted towards fixation crosses (red cross control), responses were attenuated and short, and did not entail subsequent internal reactivations (also see theories of depth of processing²⁶). Third, stereotypical internal reactivations occurred as soon as comparisons to the preferred concept had to be performed (Fig. 2a-b), even retrospectively during the question control (Fig. 6a, Supplementary Fig. 5, see also the next section), and were consistent across experimental conditions (Fig. 3c). Finally, concept neurons increased firing when their preferred concept was about to become relevant. Anticipatory single-unit activity has been reported for repeated presentations of visual stimuli in fixed sequences²⁷. In our paradigm, however, pairs of concepts were depicted randomly. While the identity of the second picture was constrained by that of the first, it could only be inferred stochastically (probability of 1/3 instead of 1/4). If the preferred concept was not shown first, it was more likely to be shown second, and firing increased shortly beforehand (Fig. 5a). This is consistent with a pre-allocation of attention for upcoming comparisons²⁸ and supports the notion that the MTL is involved in inference, planning, and imagination^{29,30}.

Reactivations of concept neurons after activity silence

Bulk imaging studies of working memory suggest that reactivations of maintained memory items can occur after periods of activity silence whose duration exceeds that of non-preferred stimulus presentations¹⁸. It has been asked whether the same applies to the firing of concept neurons with lower detection thresholds¹⁰. While true for single neurons during free recall³¹, our question comparison control sheds new light onto the reactivation dynamics of concept neurons in working memory. A concept contained in the question had to be compared separately to each picture concept before a prompt revealed which of the two comparisons had to be reported. In contrast to the main condition, first picture presentations of the preferred concept were not associated with persistent firing (Fig. 2a, green versus blue) further supporting the attention hypothesis outlined above. Specifically, attention was diverted away from the preferred concept when a second non-preferred picture had to be compared to a non-preferred concept mentioned in the initial question. Remarkably, activity then re-appeared

after complete activity silence upon presentation of the answer prompt if and only if the first comparison (between question and first preferred picture) had to be reported (Fig. 6a). Concepts were thus maintained in working memory despite activity silence of concept neurons (see also Supplementary Fig. 6).

Alternatives to persistent activity

Reactivations after activity silence point to hidden memory states residing either within cells themselves^{21,22,32} or within the network²⁰. Assuming the latter, memory items could be maintained either outside (e.g., in inferotemporal³³, perirhinal^{34,35}, parietal³⁶, or prefrontal cortex³⁷) or within the MTL in either dynamical or coherent firing, or in distributed patterns of synaptic weights²⁰. For humans, visually selective neurons in the MTL may contain information about memory items via firing at preferred phases of ongoing theta oscillations under high memory load³⁸. Supporting synaptic mechanisms, trial-relevant but unattended memory items can be reactivated after activity silence by a single pulse of TMS¹⁸. Our findings are consistent with such a state-dependent network response. Reactivations were preceded either by specific activation of other concept neurons (Fig. 5) or by increased firing of non-visually-selective neurons within the MTL (Fig. 6). Non-specific activation during instruction cues has previously been reported¹¹ and could facilitate reactivations^{18,39}. Trial-wise correlation patterns between concept neurons in our data support the idea that the presentation of non-preferred stimuli could contribute to reactivations (Fig. 5b, Supplementary Fig. 2c). Specifically, the activity of pairs of concept neurons n2 (pic2: 0-1000 ms) and n1 (pic2: 500-1300 ms) of the same hemisphere exhibited trial-wise correlations after the onset of the second picture whenever neuron n2's preferred stimulus was shown, regardless of whether the preferred stimulus of neuron n1 was shown first (reactivation) or not (Fig. 5b, dark and light blue). If neurons n2's preferred picture was not shown in second position, no pairwise correlations were present (Fig. 5b, turquoise and yellow). Correlations thus appear to depend on the preferred-stimulus responses of neuron n2, but not on the previous activation of neuron n1. Correlation patterns were thus no sufficient cause, but did plausibly contribute to reactivations, together with additional factors such as changes in excitability or intrinsic plasticity of neuron n1 due to recent activation^{22,40} (see also Supplementary Fig. 4), question-specific (top-down) input to concept neurons, or the strengthening of task-specific dynamical pathways via synaptic modifications resulting from sequential activation of (concept) neurons (Fig. 3b)."

2. Population response (Fig. 2). I have a few questions of clarification for this figure. Fig 2A: is this for all n=61 concept cells or only the 33 that showed the response? 2B: here it is clear that all n=61 are shown. I find it hard to see how robust the effect is across cells in Fig 2. It would be better to show this as a scatter to be able to appreciate the full distribution of the kinds of responses visible (showing reactivation. Also, was the 1/3rd of concept cells that showed responses to non-preferred images when related to the question the same than those that showed reactivation to pictures? (comparing 2A+B; green line in 2A shows no response?).

We apologize for the lack of clarity. For the normalized population responses in Fig. 2a, all 61 concept neuron were included, which we now explicitly state in this and all other subfigures. The green line from the question comparison condition in 2a does not show a reactivation response because whenever the first picture depicted the preferred concept, the to-be-compared concept from the question had to refer to a non-preferred concept by design (since no concept could occur more than once within the same trial). In 2b however, the preferred concept was part of the question and, consequently, reactivation responses were seen during the presentation of each of the two subsequent pictures (that did not depict the preferred concept). We now stress this design feature more clearly in the results section of the manuscript:

“Results [...]

Neither control condition required a comparison between the concepts depicted in the pictures. Either no comparisons (“no comparison” control) or comparisons of both picture concepts to a third concept mentioned in the question (“question comparison”) were required. **Since no concepts were presented more than once within the same trial, a preferred first picture (tie) in the question comparison condition meant that the second picture had to be compared to a non-preferred concept mentioned in the question.** Consequently, relational responses during second non-preferred picture presentations were abolished in both control conditions (Fig. 1d, “no comparison” and “question comparison”).”

The robustness of the effects and the relationship of reactivation responses between experimental conditions are now visualized in Fig. 2c by means of 4 new scatter plots:

Fig. 2 Relational responses of the entire population of concept neurons. **a.** Averaged normalized firing rates of concept neurons during both picture presentations in all three experimental conditions whenever the first picture represented the preferred concept. Solid lines and shaded areas depict means and standard errors, respectively. Picture on- and offsets are marked by black dashed lines. Relational responses occurred during the “reactivation window” (gray dashed lines) in the main comparison condition (blue), but not in the no- or question-comparison control condition (red and green, respectively). Time periods of significant z-value differences between main and control conditions (same colors) or zero (black) are indicated by solid lines (cluster permutation test). **b.** Heat

plot of z-values of all 61 concept neurons during the question-comparison condition sorted in descending order of activity. Dashed white lines denote onsets of different events (q: question, p1: picture 1, p2: picture 2). Top row: When the preferred concept was part of the question, responses to the question and following both non-preferred pictures were present. Bottom: When neither question nor pictures contained the preferred concept, no responses or reactivations occurred. **c. Scatter plot of mean z-values of all 61 concept neurons comparing responses and reactivations during different stimulus presentations (q: question, p1: picture 1, p2: picture 2) and experimental conditions. Subscripts indicate the experimental condition (main in blue: main experiment, q-comp in green: question comparison) while superscripts distinguish whether the preferred concept was depicted in the current (“response”: response trials 0-1000 ms after stimulus onset) or a preceding stimulus (“react.”: reactivation trials 500-1300 ms after stimulus onset). Correlation strengths and p-values for each condition pair are shown at the top and visualized by regression lines. Left two subplots: Reactivations during the main experiment predict reactivations in the question control condition ($r > 0.5$, $p < 0.0005$). Right two subplots: Response strengths to questions containing the preferred concept predict reactivation strengths during both subsequent picture presentations ($r > 0.35$, $p < 0.005$).**

The left two scatter plots in Fig. 2c now depict the relationship between reactivation responses in the main condition (picture 1 preferred as in Fig. 2a in blue) and each of the two reactivation responses in the question control condition (question preferred as in the top half of 2b) for all concept neurons. Indeed, there is a strong correlation between reactivation strengths in both experimental conditions. Moreover, the relationships of question responses and subsequent reactivations during the question-control condition are now visualized in the right two subplots of Fig. 2c. We thank the reviewer for suggesting these additional analyses that are now mentioned in the results and methods:

“Results [...]

Concept neurons indicate the presence of a relation to their preferred concept [...]

During the “question comparison” control condition, approximately one third of the concept neurons responded not only to questions containing the preferred concept but also to both subsequent non-preferred pictures (Fig. 2b, top row). Responses were absent when neither the question nor the pictures depicted the preferred concept (Fig. 2b, bottom row). **Finally, we tested whether response and reactivation patterns of concept neurons were consistent across experimental conditions. Factors predictive of reactivations in the question control condition were visualized as scatter plots of mean normalized activity for all 61 concept neurons (Fig. 2c). Both reactivation response strength to second non-preferred pictures in the main condition (Fig. 2c, left two subplots) as well as preferred question response strength (Fig. 2c, right two subplots) predicted non-preferred picture reactivation strength in the question-comparison control. Specifically, Pearson correlations between reactivations in the main condition versus relational responses in the question comparison condition to first ($r = 0.528$, $p < 0.0005$) or second non-preferred pictures ($r = 0.625$, $p < 10^{-7}$) as well as correlations between preferred question response strengths versus first ($r = 0.359$, $p < 0.005$) or second picture reactivations ($r = 0.520$, $p < 0.00005$) in the question comparison condition were highly significant.”**

“**Methods [...]**

Relational responses at the population level

[...] Responses (0-1000 ms after stimulus onset) and reactivations (500-1300 ms after stimulus onset) obtained from convolved normalized activity were compared during different stimulus presentations and experimental conditions (Fig. 2c). Pairwise relationships were depicted as scatter plots and quantified via Pearson correlation strengths visualized as regression lines.”

3. Cross-correlation analysis and claims that such interactions are local (Fig. 3). First, the claim made is that this can only be seen for local pairs. But this is not substantiated quantitatively. What does 3B look like for non-local pairs? More broadly, why would one only expect this to occur on local pairs?

We thank the reviewer for the relevant remark concerning our interaction analyses. The supplementary material now contains a figure that depicts (shift-corrected) cross correlations for non-local pairs of concept neurons (Supplementary Figure 2, see below). The overall shape of correlations remains the same for non-local pairs of concept neurons. We focused on local correlations in order to determine whether a necessary condition for the modifications of synapses is met where we deemed it most likely to occur. However, we agree that non-local correlation patterns should be taken into account and now describe them explicitly in the results section (see next point).

Second, it wasn't clear to me whether these cross-correlograms were corrected for what would be expected by covariation of firing rates by subtracting the shift predictor (for example, see Hirabayashi & Miyashita, 2005). If the shift predictor takes the same shape the interpretation of the data would rather be that the cells co-vary in firing rate due to some form of common input.

Cross correlations were not shift-corrected. Shift predictors have now been computed and subtracted according to sources mentioned in Hirabayashi & Miyashita. They can be appreciated in Supplementary Figure 2 for non-local (b) and local (a) pairs of concept neurons. While cross-correlation effects are mainly stimulus-driven, shift-corrected correlograms still exhibit peaks and differences between experimental conditions at positive lags of about 300 ms during second picture presentations for local pairs of concept neurons. These could indeed arise from (question-modulated) common input, top-down firing rate modulations, etc. We now added these analyses to the manuscript:

“**Results [...]**

Pairwise relations are revealed by the activity of pairs of concept neurons

[...]

During second picture presentations of the main condition, however, two prominent correlation peaks (around -250 ms and +350 ms) were found (Fig. 3b, bottom) and cross correlations differed significantly from control conditions on short (<25 ms) as well as longer (200 to 700 ms) timescales ($p < 0.01$; cluster permutation test, see Methods). The positive cross correlation peak corresponding to neuronal firing in the reverse order of the presentation of preferred pictures (i.e., a later reactivation of the response to the first picture) was most pronounced. **Cross correlations for non-local pairs of concept neurons exhibited the same overall pattern (Supplementary Fig. 2a). In order to disentangle stimulus-induced correlations from potential interactions between concept neurons, we subtracted non-simultaneous cross-correlograms of consecutive trials (shift predictors) from simultaneous ones for non-local (Supplementary Fig. 2b) and local (Supplementary Fig. 2c) pairs of concept neurons. After correction, only local pairs still showed a cross**

correlation peak at around +300 ms during second-picture presentations that differed significantly between experimental conditions ($p < 0.05$; cluster permutation test).”

“Methods [...]

Cross-correlations

[...]

Cross-correlations of non-local pairs of concept neurons across wire bundles were computed analogously (Supplementary Fig. 2a). In order to capture stimulus-independent interactions, shift predictors were obtained by calculating cross-correlograms from non-simultaneous pairwise activity of consecutive trials as described elsewhere⁶². After baseline normalization (geometric mean) and convolution (100 ms boxcar kernel) they were subtracted from simultaneous cross-correlograms for non-local (Supplementary Fig. 2b) and local (Supplementary Fig. 2c) pairs of concept neurons and compared between experimental conditions (cluster permutation test, $\alpha = 0.05$).”

Supplementary Figure 2. Cross-correlograms for non-local pairs (a) and after subtraction of shift-predictors (b and c). **a.** Population plots of trial-by-trial cross-correlograms between all 66 non-local pairs of concept neurons for trials in which both (non-identical) preferred concepts were depicted in either the main experiment (blue) or the control conditions (red). Means and standard errors are denoted by solid lines and shaded areas. Red horizontal lines indicate significant differences ($p < 0.05$) between the main comparison condition and controls as quantified by a cluster permutation test. **b.** Same as a but with shift correction obtained by subtracting cross-correlograms from non-simultaneous pairwise activity of consecutive trials (shift predictor). **c.** Same as b but calculated for all 22 local pairs of concept neurons from the same wire bundle.

4. Comparison of concept cell reactivation responses between brain areas. First, Fig 4A is expressed in terms of # units. But the total # units recorded in each area presumably differs. To substantiate this claim, would have to show that this also holds as a proportion of recorded neurons.

We agree that this as important distinction. Fig. 4a now depicts both absolute as well as relative numbers of units per brain region. While relative number of concept neurons were high in all recorded MTL areas (about 2%), the ratio of concept to visually selective neurons was lowest in PHC, and reactivation responses still most likely occurred in EC, H and A (about 3 percent). The results section now reads:

“Results [...]

Representations of abstract relations are associated with local correlations

While concept neurons were found in all brain regions of the medial temporal lobe (approximately 2% of units; Fig. 4a, top), their proportion with respect to visually selective neurons was lowest in parahippocampal cortex. Furthermore, relational responses (reactivated neurons) were most frequent in areas associated with declarative memory function, namely in amygdala, hippocampus and entorhinal cortex (approximately 3% of units; Fig. 4a, bottom). Visual neurons, i.e. neurons that responded only to the presented stimulus, but not to the compared concepts (no relational responses), on the other hand, were most frequent in parahippocampal cortex.”

Fig. 4 Relational responses occur in brain regions associated with episodic memory and require conceptual processing. **a. Relative frequency of each neuron type expressed as percentage of recorded units for different brain regions (PHC: parahippocampal cortex, EC: entorhinal cortex, H: hippocampus, A: amygdala).** Top: Visually selective neurons are divided into “concept” and non-concept (“visually selective”) neurons. Bottom: Concept neurons are split into “reactivated neurons” with relational responses and remaining “input-driven neurons”. **b.** Boxplots of z-values (Q1, median, Q3; whisker: points within +/- 1.5 IQR) of average normalized relational response activity during the reactivation window of the main condition for merely visually selective versus actual concept neurons. Trials requiring semantic processing (“Bigger?”, “Last seen in real life?”, “More expensive/Older?”) are colored in light blue, those depicting the perceptual question (“Brighter?”) in dark blue. Only relational responses of concept neurons during semantic questions deviated significantly from baseline. Brackets with asterisks show results of pairwise Mann-Whitney U tests; ****, $p < 0.0001$; ***, $p < 0.001$; **, $p < 0.01$ (uncorrected).

Second, some more details need to be given for this analysis. For Fig 5B, how many neurons are included (i.e. only the 33 reactivated or all 61)? Also is what is plotted only for the preferred stimulus? If so, is there an effect on the non-preferred responses?

All 61 concept neurons were candidates for pair-wise correlation analyses. Concept neuron pairs (n2 and n1) with non-identical preferred concepts from the same hemisphere of the same session (from different channels) were analyzed with respect to their activity during trials in which each of their preferred concept was shown (response and reactivation). 22 of such pairs were contained within and 18 pairs across wire bundles. We now distinguish four cases, depending on whether each neuron's preferred concept was shown or not, in an extended version of Fig. 5b. The activity of pairs of concept neuron n2 (pic2: 0-1000 ms) and n1 (pic2: 500-1300 ms) of the same hemisphere exhibited trial-wise correlations after the onset of the second picture whenever neuron n2's preferred stimulus was shown regardless of whether the preferred concept of neuron n1 was shown first (reactivation) or not at all (Fig. 5b, dark and light blue). If neurons n2's preferred picture was not shown second, no pairwise correlations were present (Fig. 5b, turquoise and yellow). Correlations thus appear to depend on preferred stimulus responses of neuron n2 but not on the activation history of neuron n1. They could thus reflect a necessary but not a sufficient condition for the reactivation of neuron n1. Additional factors contributing to reactivations could include the immediate activation history of the to-be-reactivated neuron (as the reviewer rightly suggested below) as well as question-specific input to concept neurons since reactivations only occurred when a comparison to the preferred concept was suggested by the question. Since the activation history of neuron n1 could affect its excitability (Pignatelli et al., 2019) or intrinsic plasticity (Zhang et al., 2003) and thereby contribute to reactivations, we analyzed the relationship between preceding responses and reactivations in Supplementary Figure 4. Previous response strengths were correlated with reactivation strengths both on a population and on a trial level (see also Supplementary Figure 4 below).

We added a description of these findings to the results section and address them in the discussion:

“Results [...]

For each pair, the response strength of one neuron (n2) to its preferred concept presented in second position was compared to the reactivated response of the other neuron (n1) whose preferred stimulus had been shown in first position. Correlations were positive and significantly different from zero for pairs within, but not across wire bundles of the same hemisphere ($p=0.022$ vs. $p=0.983$; right-tailed Wilcoxon signed-rank test, Fig. 5b, **boxplots in dark blue**). **Similarly, pairwise normalized firing rates of all response-reactivation trials of these pairs were correlated strongly within ($p<0.0001$; Pearson correlation), but not across wire bundles ($p=0.981$; Fig. 5b, scatter plots in dark blue). Correlation strengths did not differ significantly between perceptual and semantic trials ($p=0.089$ within, $p=0.319$ across wire bundles; Mann-Whitney U test; Supplementary Fig. 3b). During trials in which neither preferred concept was shown, on the other hand, pairwise correlations did not exceed chance, neither within ($p=0.168$) nor across ($p=0.071$) wire bundles (Fig. 5b, boxplots in yellow). Remarkably, correlations within bundles were always significant if the second picture showed the preferred concept of neuron n2, even if the first picture did not depict the preferred concept of neuron n1 (light blue, $p<0.001$ within, $p=0.327$ across bundles). During non-preferred second picture presentations of neuron n2, no significant correlations could be detected, even for reactivation trials of neuron n1 (turquoise, $p=0.306$ within, $p=0.948$ across bundles).”**

“Discussion [...]

Alternatives to persistent activity [...]

Trial-wise correlation patterns between concept neurons in our data support the idea that the presentation of non-preferred stimuli could contribute to reactivations (Fig. 5b, Supplementary Fig. 2c). Specifically, the activity of pairs of concept neurons n2 (pic2: 0-1000 ms) and n1 (pic2: 500-1300 ms) of the same hemisphere exhibited trial-wise correlations after the onset of the second picture whenever neuron n2’s preferred stimulus was shown, regardless of whether the preferred stimulus of neuron n1 was shown first (reactivation) or not (Fig. 5b, dark and light blue). If neurons n2’s preferred picture was not shown in second position, no pairwise correlations were present (Fig. 5b, turquoise and yellow). Correlations thus appear to depend on the preferred-stimulus responses of neuron n2, but not on the recent activation of neuron n1. Correlation patterns were thus no sufficient cause, but did plausibly contribute to reactivations, together with additional factors such as changes in excitability or intrinsic plasticity of neuron n1 due to recent activation^{22,40} (see also Supplementary Fig. 4), question-specific (top-down) input to concept neurons, or the strengthening of task-specific dynamical pathways via synaptic modifications resulting from sequential activation of (concept) neurons (Fig. 3b).”

Fig. 5 [...] b. Pairwise activity of concept neurons n2 and n1 with non-identical preferred stimuli during the main experiment in either the response (n2: 0-1000 ms) or the reactivation window (n1: 500-1300 ms) of picture 2. Pairs from the same hemisphere but different microwires were obtained from within (N=22, left) or across (N=18, right) wire bundles. Boxplots (Q1, median, Q3; whisker: points within +/- 1.5 IQR) of pairwise Spearman correlation effect sizes (rho) for different trials distinguished by color. Dark blue: both pictures preferred (n1: 1st picture, n2: 2nd picture). Yellow: none of the pictures preferred. Inlets contain scatter plots and regression lines of normalized pairwise activity of trials when both pictures were preferred (dark blue). P-values and effect size of regressions are shown on top (left: $r=0.17$, $p<0.001$; right: $r=0$, $p=0.98$).

Supplementary Figure 4. Response strengths to preferred concepts shown first predict reactivation strengths both on a population (a) and on a trial level (b), pointing to potential differences in excitability or intrinsic plasticity due to previous activation. **a.** Scatter plot of mean normalized population activity of concept neurons between preferred picture responses ($p1^{\text{response}}$: picture 1 response, 0-1000 ms) and reactivations ($p2^{\text{react}}$: picture 2 reactivation, 500-1300 ms) from reactivation trials of the main experiment as denoted by the subscript. Linear fit results (top right corner) with outlier correction (Iteratively Reweighted Least Squares) are visualized by regression lines. Population responses to first preferred pictures significantly correlate with population reactivation strengths ($r=0.21$, $p<10^{-9}$). **b.** Distribution of trial-wise correlation effect sizes between either baseline (-400-100 ms, left), early (100-600 ms, middle) or late (600-1100 ms, right) response activity to the preferred picture in first position ($p1$ resp.) and reactivations ($p2$ react.: 500-1300 ms). Both early ($z=3.423$, $p<10^{-3}$; Wilcoxon signed-rank test) and late responses ($z=2.898$, $p=0.004$), but not baseline activities ($z=1.555$, $p=0.120$) predict trial-wise fluctuations in reactivation strengths.

5. Claim of local interactions (Fig 5B). First, the finding of enhanced responses to preferred stimuli if they were shown in the **second position only during the main condition (Fig 5A,C)** is very interesting. Was this seen during both the semantic and the perceptual comparison questions or was this also related to the semantic question only like in Fig 4B?

Enhanced responses in the second position for perceptual versus semantic trials are now examined in Supplementary Figure 3. It depicts the positional response strength difference over time. While positional differences were indeed more pronounced during semantic trials, these differences did not reach statistical significance. This is now described in the results section:

“**Results** [...]

Representations of abstract relations are associated with local correlations

[...] Normalized relational responses (500 to 1300 ms) only differed significantly from zero when the question required semantic processing of the pictures (“**semantic trials**”, $p=1.74*10^{-7}$; Wilcoxon signed-rank test, Fig. 4b). Perceptual processing (“perceptual trials”, i.e. “Brighter?”) for concept neurons or visual selectivity alone (“visual neurons”, both perceptual and semantic trials) was associated with significantly lower normalized activity (all three $p<0.005$; Mann-Whitney U test) not significantly different from zero (Wilcoxon signed-rank test). Moreover, concept neurons responded more strongly to their preferred concept in second versus first picture positions, particularly during the early response phase immediately

preceding the period of relational responses (“reactivation window”; Fig. 5a, $p < 0.01$; cluster permutation test) and even before the onset of the second picture (“anticipation window”; Fig. 5a). **Despite being more pronounced in semantic trials, positional response strength differences did not differ significantly from perceptual trials (Supplementary Fig. 3a; $p = 0.13$; Wilcoxon signed-rank test).”**

Supplementary Figure 3. Positional response strength differences and pairwise correlations did not differ significantly between perceptual and semantic trials. **a.** Normalized positional difference of concept-neuron responses ($N=61$) to the preferred concept in second minus first picture position for perceptual (dark blue) versus semantic (blue) trials. Positional differences were slightly more pronounced in semantic trials without reaching significance (Wilcoxon signed-rank test, 100-600 ms, $p=0.13$). **b.** Effect sizes of Spearman correlations of firing rates for pairs of concept neurons in semantic (blue) versus perceptual (dark blue) trials. Activity from each pair of neurons n_2 and n_1 was obtained during the second response (n_2 : 0-1000 ms) or reactivation window (n_1 : 500-1300 ms) and from trials in which both (non-identical) preferred concepts were shown (first that of n_1 , then that of n_2). Effect sizes of pairwise correlations were not significantly different in semantic versus perceptual trials, neither within nor across bundles of the same hemisphere (Mann-Whitney U test within: $p=0.09$, across: $p=0.32$)

Second, Fig 5B makes the claim that the extent of the reactivation response to non-preferred stimuli is positively correlated with the response in the same trial of concept cells that prefer the second stimulus. This striking phenomena deserves more detailed analysis (scatter, or PSTH).

Figure 5b (see above) now includes scatter plots of the normalized activity of all pairs of concept neurons for trials in which their respective preferred concepts were shown (see above). They are depicted as inlets: one inlet for pairs within (left) and one for pairs across (right) wire bundles:

“Results [...]

For each pair, the response strength of one neuron (n_2) to its preferred concept presented in second position was compared to the reactivated response of the other neuron (n_1) whose preferred stimulus had been shown in first position. Correlations were positive and significantly different from zero for pairs within, but not across wire bundles of the same hemisphere ($p=0.022$ vs. $p=0.983$; right-tailed Wilcoxon signed-rank test, Fig. 5b, **boxplots in dark blue**). **Similarly, pairwise normalized firing rates of all response-reactivation trials**

of these pairs were correlated strongly within ($p < 0.0001$; Pearson correlation), but not across wire bundles ($p = 0.981$; Fig. 5b, scatter plots in dark blue).”

Is there temporal overlap between the two responses or is there a ‘silent’ delay in between?

This is an important question. The exact temporal relationship between responses to preferred pictures shown second and reactivation responses can now be appreciated in Fig. 5a (left side). During the early response phase up to about 500 ms, there is indeed no overlap. In other words, initially to-be reactivated neurons are still silent (compatible with shifted cross correlation peaks in Fig. 3b and in the additional correlation analyses of Supplementary Figure 2). It is only later that reactivation responses overlap with the late preferred stimulus responses (see also next point).

Fig. 5 Relational responses could arise from local interactions of concept neurons. **a. Left:** Averaged normalized firing rates of concept neurons with standard errors (shaded areas) when either the first (blue) or second (red) picture was the preferred stimulus. Time periods of significant differences are indicated by solid lines in respective colors (cluster permutation test, black: zero). When the preferred concept was shown first, neural activity increased before the onset of the second picture (“anticipation window”, gray dashed lines). Second responses (red) were stronger, and their late phase coincided with reactivations of units responsive to the first picture (“reactivation window”, blue dashed lines). **Right:** Direct comparison of positional differences (cluster permutation test, periods of significant difference marked in dark gray). [...]

If there is no delay I cannot see how the claim of ‘activity-silent synaptic mechanisms’ (discussion) can be supported. A more straightforward explanation would be that one neuron excites the other, and that this excitation is somehow gated by this neuron having been active shortly before (first stimulus)? Perhaps this is what is meant, but this should be described more carefully.

We thank the reviewer for raising this important point. The activation history is now mentioned as a potential contributing factor (see above, Supplementary Figure 4) and we address the question of reactivations after activity silence in additional analyses (new main Figure 6 and Supplementary Figure 6). First, while some concept neurons exhibited sustained firing after first preferred-picture presentations with only brief activity silence during second, non-preferred pictures, others are

reactivated after longer periods of activity silence without sustained activation (Supplementary Figure 6b). Second, our question comparison condition revealed that reactivations could occur after complete activity silence upon presentation of the answer cue when the first picture showed the preferred concept. The answer cue at the end of each trial either asked for a comparison between the concept of the question (non-preferred) and either that of the first (preferred, green) or that of the second picture (non-preferred, violet). Normalized activity differed significantly between these two instructions (Fig. 6a, cluster permutation test, black) and concept neurons were reactivated only if the answer referred back to the preferred concept (first picture, green line). Following complete activity silence, reactivations began 500 ms after the onset of the answer cue. Importantly, we identified a new factor that could contribute to reactivations of concept neurons. A subset of non-visually selective neurons (no picture response in bin-wise signed-rank test) whose normalized firing relative to baseline (-400 to 100ms) exceeded a z-value of one during the first 1500 ms of the answer cue presentation, sharply increased firing about 250 ms earlier than reactivations of concept neurons (Fig. 6b). We now describe and interpret these new findings:

“Results [...]

Reactivations after activity silence following non-specific activation

Finally, we asked whether and how reactivations could occur after longer periods of activity silence. In our question comparison condition, a concept contained in the question had to be compared to that of either the first or the second picture. The answer cue at the end of each trial revealed which comparison was to be made. If the first picture depicted the preferred concept, activity of concept neurons during the answer cue differed significantly between these two alternatives ($p < 0.05$, Fig. 6a). Following complete activity silence, concept neurons were reactivated 500 ms after presentation of the answer cue but only if it referred back to their preferred concept. Meanwhile, 104 non-visually-selective neurons not responsive to any of the four pictures used per session (bin-wise signed-rank test) whose mean z-value exceeded one during the first 1500 ms (baseline -400 to 100 ms) sharply increased firing approximately 250 ms earlier than reactivated concept neurons (Fig. 6b).“

“Discussion [...]

Reactivations of concept neurons after activity silence

Bulk imaging studies of working memory suggest that reactivations of maintained memory items can occur after periods of activity silence whose duration exceeds that of non-preferred stimulus presentations¹⁸. It has been asked whether the same applies to the firing of concept neurons with lower detection thresholds¹⁰. While true for single neurons during free recall³¹, our question comparison control sheds new light onto the reactivation dynamics of concept neurons in working memory. A concept contained in the question had to be compared separately to each picture concept before a prompt revealed which of the two comparisons had to be reported. In contrast to the main condition, first picture presentations of the preferred concept were not associated with persistent firing (Fig. 2a, green versus blue) further supporting the attention hypothesis outlined above. Specifically, attention was diverted away from the preferred concept when a second non-preferred picture had to be compared to a non-preferred concept mentioned in the initial question. Remarkably, activity then re-appeared after complete activity silence upon presentation of the answer prompt if and only if the first comparison (between question and first preferred picture) had to be reported (Fig. 6a). Concepts were thus maintained in working memory despite activity silence of concept neurons (see also Supplementary Fig. 6).

Alternatives to persistent activity

Reactivations after activity silence point to hidden memory states residing either within cells themselves^{21,22,32} or within the network²⁰. Assuming the latter, memory items could be maintained either outside (e.g., in inferotemporal³³, perirhinal^{34,35}, parietal³⁶, or prefrontal cortex³⁷) or within the MTL in either dynamical or coherent firing, or in distributed patterns of synaptic weights²⁰. For humans, visually selective neurons in the MTL may contain information about memory items via firing at preferred phases of ongoing theta oscillations under high memory load³⁸. Supporting synaptic mechanisms, trial-relevant but unattended memory items can be reactivated after activity silence by a single pulse of TMS¹⁸. Our findings are consistent with such a state-dependent network response. Reactivations were preceded either by specific activation of other concept neurons (Fig. 5) or by increased firing of non-visually-selective neurons within the MTL (Fig. 6). Non-specific activation during instruction cues has previously been reported¹¹ and could facilitate reactivations^{18,39}. [...]"

“Methods [...]

Definition of unit types

[...] Non-visually-selective neurons as defined in Fig. 6 did not respond to any picture (bin-wise signed-rank test) but instead exhibited normalized firing relative to baseline (-400 to 100 ms) exceeding one during the 1500 ms upon presentation of the answer cue.”

Fig. 6 Concept neurons are reactivated after activity silence (a) following non-specific activation (b) whenever attention is directed back towards their preferred concept. **a.** Averaged normalized firing rates of all concept neurons with standard errors (shaded areas) in the question comparison condition during the presentation of the answer prompt at the end of each trial whenever the first picture was preferred. The answer cue asked for a comparison between the concept of the question (non-preferred) and either that of the first (preferred, green) or that of the second picture (non-preferred, violet). Normalized activity differed significantly between these two answer prompts (cluster permutation test, black) and concept neurons were reactivated only if the answer referred back to the preferred concept (green). Following complete activity silence, reactivations began 500 ms after presentation of the answer cue (see also reactivation window in Fig. 5a). **b.** Averaged normalized firing rates of non-visually-selective neurons whose z-value exceeded one during the answer cue (0-1500 ms). Blue and red again denote comparisons to first (blue) or second (red) picture concepts. Activity sharply increased after approximately 250 ms, roughly 250 ms before reactivations of concept neurons, and did not differ between the two answer prompts.

Supplementary Figure 6. Persistence of firing varies between concept neurons **a.** Left: Heat plot of z-values of a concept neuron in the left hippocampus across trials in which the preferred concept was depicted first (main comparison condition). Dashed white lines denote onsets and offsets of picture presentations (p1: picture 1, p2: picture 2). After the response to the first, preferred picture, firing was increased for almost the entire duration of the trial with the exception of a short period of activity silence during the presentation of the second, non-preferred pictures. Right: Corresponding averaged normalized firing rates. Solid lines and shaded areas depict means and standard errors, respectively. **b.** Same as **a**, but for a different concept neuron in the right hippocampus of the same patient, but from a different experimental session. Here, after a short response to the preferred picture in first position, activity was not sustained (if at all with brief rhythmic bouts of activity), yet still reactivated in the later phase of the presentation of the non-preferred pictures in second position after a longer period of activity silence.

Also, the claim is advanced that ‘reactivation’ only happens during the semantic trials of the main condition. How is this taken into account for Fig 5B, i.e. since there is no reactivation during perceptual, there should also be no such correlation in the local condition?

Boxplots of pairwise correlation strengths distinguishing perceptual from semantic trials have been added to the supplement in Supplementary Figure 3b. While correlations strengths were indistinguishable between trial types (and even higher during perceptual trials), correlations strengths could only be computed from 5 trials per neuron pair for the perceptual condition and should thus be interpreted with caution. Furthermore, additional valuable analyses suggested by the reviewer revealed that trial-wise noise correlations between pairs of concept neurons alone are not sufficient to cause reactivations (see above, Fig. 5b). Reactivation differences between perceptual and semantic trials could thus arise from factors independent of noise correlations, such as question-specific input to concept neurons. These considerations are now part of the discussion (see above) and the following sentence has been added to the results:

“Results [...]

Correlation strengths did not differ significantly between perceptual and semantic trials ($p=0.089$ within, $p=0.319$ across wire bundles; Mann-Whitney U test; Supplementary Fig. 3b).”

Minor issues:

1. Definition of ‘concept neurons’. The definition of a concept neuron here is one that responds vs. baseline selectively to only a given picture as well as also to instructions (text) that feature that object. It would be useful to compare this definition with others used in the literature so far (i.e. multimodal? Different images to same object).

We added a respective paragraph to the discussion.

“Discussion [...]

On the other hand, while responses to concepts were selective and invariant, responding to both pictures and text and thus likely multimodal³⁶, it is unclear whether information about concepts per se is encoded.”

2. concept cell firing to different photographs involving the concept ‘airplane’.

Unfortunately, we are not sure what reviewer means. We checked our screening sessions and found that no airplane was included in any of them.

3. In the introduction the authors seem to be downplaying the current state of the field with regard to relational processing in the hippocampus. While such task dependent relational firing has not been demonstrated prior, statements like “It is currently not known however, whether or how concept neurons dynamically encode conceptual relations and could thereby account for the integrative properties of episodic memories” ignores the fact that it is known that hippocampal concept cells fire to related concepts. Papers like Ison et al 2015, Staresina et al 2019 (both of which the authors cite later), and Rey et al 2018 are good examples of human single unit papers that demonstrate this.

Evidence for firing to related concepts is now cited as suggested in the revised version of the introduction. The passage in question now reads:

“Introduction [...]

Visually selective neurons in the human MTL represent preferred visual stimuli in working memory through persistent firing¹¹⁻¹³ that predicts memory performance^{11,12}

and reflects associated stimuli¹⁴⁻¹⁶. How concept neurons dynamically encode conceptual relations and account for their storage, however, is not completely understood.“

4. *For many figures it is not clear to me what the n is. This should be stated or individual datapoints shown (i.e. Figure 4b, 5b, 5c).*

All Ns have been added to the figures and data points can now be appreciated as scatter plots.

5. *Line 167 – conceptual processing needs to be defined – how is this different from semantic processing?*

We now state more explicitly at the beginning of the discussion section what was meant by conceptual processing:

“Importantly, reactivations depended on the explicit recognition and further processing of concepts.”

6. *Line 189-190 – “On the other hand, relational signals likely enter the hippocampus when episodic memories are formed” This needs a citation.*

This sentence is no longer part of the revised discussion.

7. *Line 221-223 – This sentence “...neural drift” needs a citation.*

This sentence is no longer part of the revised discussion.

REVIEWERS' COMMENTS

Reviewer #1 (Remarks to the Author):

The authors have done an impressive job dealing with the comments made by both reviewers. I am pleased to say that, for my part, I think that their reactions have substantially improved the manuscript. As a result, I am happy to say that they have removed the reservations that I had when reading the original version.

Reviewer #2 (Remarks to the Author):

The authors prepared an extensive revision, with rigorous and detailed attention to all the issues I had raised. The new figures and analysis added conclusively address the questions I had offered, and indeed offer deep new insight the original version did not have.

I would like to congratulate the authors on a very impressive project and paper, which I anticipate will have a very high impact on the field. This paper is in my opinion now very strong and ready for publication in Nature Communication as-is.

(very) Minor issues:

Fig 2c – it wasn't clear to me which of the two conditions plotted in each scatter is on x axis and which on y axis, please label.

Fig 6, x axis label I found a bit confusing ("answer" ?). I assume the authors mean "relative to onset of question" (i.e. the question screen in Fig

Reviewer #1 (Remarks to the Author):

The authors have done an impressive job dealing with the comments made by both reviewers. I am pleased to say that, for my part, I think that their reactions have substantially improved the manuscript. As a result, I am happy to say that they have removed the reservations that I had when reading the original version.

We thank the reviewer for important contributions to the improved manuscript and are grateful for the constructive feedback.

Reviewer #2 (Remarks to the Author):

The authors prepared an extensive revision, with rigorous and detailed attention to all the issues I had raised. The new figures and analysis added conclusively address the questions I had offered, and indeed offer deep new insight the original version did not have.

I would like to congratulate the authors on a very impressive project and paper, which I anticipate will have a very high impact on the field. This paper is in my opinion now very strong and ready for publication in Nature Communication as-is.

We are pleased with the reviewer's assessment of the relevance and future impact of this project. Again, we wish to express gratitude for the reviewer's constructive feedback and substantial contributions.

1.

(very) Minor issues:

Fig 2c – it wasn't clear to me which of the two conditions plotted in each scatter is on x axis and which on y axis, please label.

We now state which condition corresponds to the x and y axis, respectively:

2.

Fig 6, x axis label I found a bit confusing ("answer" ?). I assume the authors mean "relative to onset of question" (i.e. the question screen in Fig

The x-label was indeed a bit confusing, given that in figure 1b the "answer prompt" indeed poses a question. Therefore, we changed the x-label in Fig 6. to "time relative to response prompt (ms)" in order to distinguish this response-question screen at the end of the question-control condition from the question screen at the beginning of the trial in the main experiment.

Fig. 6 Concept neurons are reactivated after activity silence (a) following non-specific activation (b) whenever attention is directed back towards their preferred concept. **a.** Averaged normalized firing rates of all concept neurons with standard errors (shaded areas) in the question-comparison condition during the presentation of the **response prompt** at the end of each trial (see Fig.1b) whenever the first picture was preferred. The **response prompt** (onset: vertical dashed line in dark red) asked for a comparison between the concept of the question (non-preferred) and either that of the first (preferred, green) or that of the second picture (non-preferred, violet). Normalized activity differed significantly between these two **response prompts** ($p < 0.01$; two-sided cluster permutation test, black) and concept neurons were reactivated only if the **prompt** referred back to the preferred concept (green). Following complete activity silence, reactivations began 500 ms after presentation of the **response prompt** (see also reactivation window in Fig. 5a). **b.** Same as a but with averaged normalized firing rates of non-visually-selective neurons whose z -value exceeded one during the **response prompt** (0-1500 ms; onset: vertical dashed line in dark red). Blue and red again denote comparisons to first (blue) or second (red) picture concepts. Activity sharply increased after approximately 250 ms, roughly 250 ms before reactivations of concept neurons, and did not differ between the two **response prompts** ($p < 0.01$; two-sided cluster permutation test). Data are presented as mean values \pm SEM (a, b) with solid lines and shaded areas, respectively. Source data are provided as a Source Data file.